# In vivo Hox binding specificity revealed by systematic changes to a single *cis* regulatory module

Carlos Sánchez-Higueras [1], Chaitanya Rastogi [2], Roumen Voutev [3], Harmen J. Bussemaker [2], Richard S. Mann [3] & James C.-G. Hombría [1]

Hox proteins belong to a family of transcription factors with similar DNA binding specificities that control animal differentiation along the antero-posterior body axis. Hox proteins are expressed in partially overlapping regions where each one is responsible for the formation of particular organs and structures through the regulation of specific direct downstream targets. Thus, explaining how each Hox protein can selectively control its direct targets from those of another Hox protein is fundamental to understand animal development. Here we analyse a *cis* regulatory module directly regulated by seven different *Drosophila* Hox proteins and uncover how different Hox class proteins differentially control its expression. We find that regulation by one or another Hox protein depends on the combination of three modes: Hox-cofactor dependent DNA-binding specificity; Hox-monomer binding sites; and interaction with positive and negative Hox-collaborator proteins. Additionally, we find that similar regulation can be achieved by *Amphioxus* orthologs, suggesting these three mechanisms are conserved from insects to chordates.

[1] Centro Andaluz de Biología del Desarrollo (CSIC/JA/UPO), Universidad Pablo de Olavide, 41013 Seville, Spain. [2] Departments of Biological Sciences and Systems Biology, Columbia University, New York, NY 10027, USA. [3] Departments of Biochemistry and Molecular Biophysics, Systems Biology, and Neuroscience, Mortimer B. Zuckerman Mind Brain Behavior Institute, Columbia University, New York, NY 10027, USA. Correspondence and requests for materials should be addressed to C.S.-H. (email: csanhig@upo.es) or to R.S.M. (email: rsm10@columbia.edu) or to J.-G.H. (email: jcashom@upo.es)

The family of Hox transcription factors controls the diversification of an animal's morphology along the antero-posterior axis[1–4]. Hox proteins function by activating or repressing downstream target genes responsible for the formation of specific organs and unique morphological structures[5,6]. Key to Hox function is their deployment in partially complementary patterns, with each Hox protein imposing specific characteristics on the cells where it is expressed. Hox genes originated early during animal evolution and fall into several homology groups that can be identified from vertebrates to invertebrates. In *Drosophila* there are eight Hox genes that expressed from cephalic to caudal segments are *labial* (*lab*), *proboscipaedia* (*pb*), *Deformed* (*Dfd*), *Sex combs reduced* (*Scr*), *Antennapedia* (*Antp*), *Ultrabithorax* (*Ubx*), *abdominal-A* (*abd-A*) and *Abdominal-B* (*Abd-B*) (Fig. 1a–c).

To identify direct Hox targets, a considerable effort has been dedicated to characterise the DNA binding sites recognised by each Hox protein. Surprisingly, in vitro analyses revealed that all Hox proteins recognise a short 6 bp AT rich sequence that, on average, is found multiple times in all genes in the genome[7]. Moreover, with the exception of Abd-B, which has one of the most divergent homeodomains and a preference for TTAT or TTAG sequences[8], all Hox proteins bind the same TAAT core sequence[7]. Thus, the observed in vitro Hox-monomer DNA-binding preferences are unable to explain the in vivo regulatory specificity imposed by each Hox protein.

The above paradox was partly resolved by the observation that besides binding DNA as monomers, Hox proteins bind with higher affinity to DNA when forming complexes with two TALE homeodomain cofactor proteins conserved in vertebrates (Pbx and Meis) and invertebrates [Extradenticle (Exd) and Homothorax (Hth)][9,10]. The formation of a Hox-Exd-Hth complex not only extends the DNA recognition site to about 12 base pairs but, more importantly, uncovers a latent specificity in each Hox protein that allows the recognition of different DNA sequences[11]. In a SELEX-seq high-throughput analysis of all eight *Drosophila* Hox-cofactor protein complexes, it was observed that Hox proteins can be classified in three groups according to their DNA binding preferences[11]. Hox class 1, comprising Lab and Pb, prefer binding to a TGATTGAT core sequence; class 2, comprising Dfd and Scr, prefer TGATTAAT; class 3, comprising Antp, Ubx, Abd-A and Abd-B, prefer TGATTTAT. Moreover, base pairs that flank these core 8mers can dramatically influence the in vitro affinity of each Hox-cofactor complex (Fig. 1k). SELEX-seq data have been used to predict in silico the existence and relative affinity of putative binding sites, but the number of predictions that have been tested in vivo is very limited[12,13].

To systematically test the idea that diverse Hox-cofactor complex affinities and Hox-monomer DNA binding sites control in vivo Hox specificity, we analysed the *Drosophila* vvl1+2 enhancer, which is regulated by most Hox genes[14]. vvl1+2 is a transient *cis*-regulatory module (CRM) responsible for the early activation of the *ventral veinless* (*vvl*) gene in a segmentally repeated pattern of patches that extend from the maxillary (mx) segment to the ninth abdominal segment (A9) on the lateral ectoderm of the *Drosophila* embryo (Fig. 1l). In segments T2 to A8 these homologous patches label the primordia of the respiratory tracheal system, while in the maxilla and labium (lb) they label the primordia of the *corpora allata* and the prothoracic glands, respectively[14]. Thus vvl1+2 is expressed in the domain of all Hox genes except for *lab* (Fig. 1a–c).

Besides Hox gene function, the correct regulation of vvl1+2 expression requires the WNT and the JAK/STAT signalling pathways. In *wingless* (*wg*) mutant embryos the repeated patches of expression are replaced by a continuous lateral stripe, suggesting that the WNT pathway acts in all segments as a negative

regulator of vvl1+2[15,16]. On the other hand, mutants blocking JAK/STAT signalling result in the almost complete disappearance of vvl1+2 expression. Mutation of the STAT binding sites in the CRM reduce its expression demonstrating STAT is a direct vvl1+2 activator[16]. Although the expression of vvl1+2 in most Hox gene domains initially obscured it as a downstream Hox target, analysis of vvl1+2 expression in either *Dfd Scr* or in *Scr Antp* double mutants demonstrated the requirement of these Hox proteins for vvl1+2 activation in their respective expression domains. Moreover, the lack of expression of vvl1+2 in *Dfd Scr* double mutants, could be rescued by activation of either Dfd, Scr, Antp, Ubx or Abd-B indicating that this CRM can be activated by most Hox proteins[14].The nearly pan-Hox regulation of the vvl1+2 enhancer differs from previously studied Hox target CRMs that are controlled in a segment-specific pattern by only one or a few Hox proteins[6,17,18].

To understand how Hox proteins that diverged so early during animal evolution can activate vvl1+2 in homologous cells of different segments, here we combine in silico, biochemical and genetic analysis in embryos. Our results demonstrate a striking correspondence between predicted in vitro affinity and in vivo activity for conferring Hox specificity. We also find that Hox-monomer binding sites can achieve the same in vivo activation as a single Hox-cofactor site, and that the role played by Hox-monomer sites is more important for trunk (class 3) than for cephalic Hox proteins (class 1 and 2). In addition to antero-posterior segmental specificity imposed by Hox-monomer and Hox-cofactor binding sites, we also uncover how the fine intra-segmental pattern driven by vvl1+2 requires positive JAK/STAT and negative WNT pathway inputs, underscoring an important role for Hox-collaborators in enhancer activity. Finally, we show that the highly divergent *Amphioxus* chordate Hox proteins can substitute for their orthologous *Drosophila* proteins. These results indicate that the regulatory rules controlling downstream target regulation in *Drosophila* are also applicable to chordate Hox proteins, providing important insights into the evolution of Hox protein function in general. By systematically modifying the nucleotide sequence of a direct Hox target, our study allows a direct comparison of Hox specificity in vivo with great precision.

## Results

**Localisation of putative Hox and Hox-cofactor binding sites.** To analyse Hox regulation of vvl1+2 we used the algorithm *No Read Left Behind* [NLRB][12] to perform an in silico search for Hox-monomer and Hox-cofactor binding sites selecting a representative protein of each class: Lab (class 1), Dfd (class 2), and Ubx (class 3). This search identified a small number of sites predicted to bind with different affinity to the three Hox-cofactor complexes (Fig. 1d–f), as well as a large number of sites predicted to bind Hox proteins as monomers (Fig. 1g–i). Monomer binding sites locate along the three different regions in which the vvl1+2 enhancer has previously been subdivided, named from distal to proximal to the promoter as *S1, S2* and *S3* [Fig. 1j and ref. [16]]. Monomer Ubx binding is predicted to be more prevalent on the *S2* and *S3* fragments than on *S1* (Fig.1i).

The affinities predicted by *NRLB* models[12] are reported as normalised relative to the genomic maximum, which corresponds to a different and unknown dissociation constant ($K_d$) for each factor. Comparing affinity scores for the same DNA sequence between factors is therefore not meaningful. However, we were able to estimate the relative contribution of Hox-monomer versus Hox-cofactor sites to the CRM's regulation by comparing vvl1+2 expression in wild type embryos with embryos mutant for the strong *homothorax[P2]* (*hth[P2]*) allele where Exd is not imported into the nuclei, effectively behaving as an *exd* mutant[9,19]. In *hth[P2]*

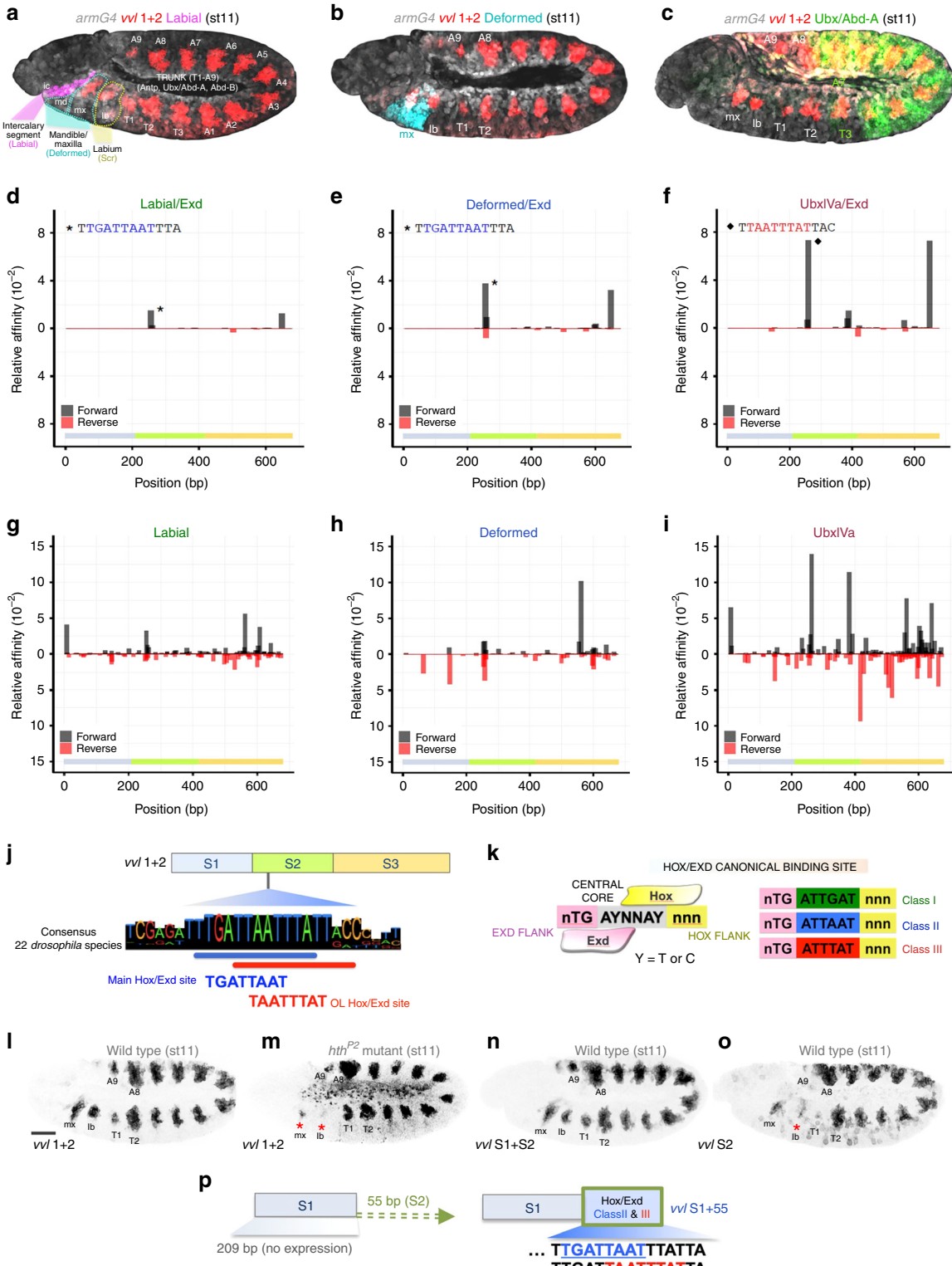

**Fig. 1** Hox regulatory inputs in *vvl1+2* CRM. **a–c** Spatial relation of *vvl1+2* expression pattern (red) with respect to various Hox expression domains. **a** Labial (purple) is expressed in the intercalary segment, (**b**) Dfd (blue) is expressed in the mandible and maxilla, (**c**) Ubx and Abd-A (green) protein expression extends from T3 to A7. **d–i** In silico predicted relative affinity and localisation of Hox binding sites for Lab-Exd (**d**), Dfd-Exd (**e**), Ubx IVa-Exd (**f**), Lab monomer (**g**), Dfd monomer (**h**) or Ubx IVa isoform monomer (**i**) in *vvl1+2*. (**j**) Scheme of *vvl1+2* and the *S1*, *S2* and *S3* sub-fragments showing the localisation of the *main* (blue) and *overlapping* (red) Hox-Exd sites. (**k**) Scheme of a canonical Hox-Exd binding site showing the central core bound by both proteins (grey), the Exd (pink) or Hox (yellow) bound flanking sites, and the core sequences bound preferentially by class 1, 2 and 3 complexes. Expression of *vvl1+2* in wild type (**l**) or *hth*[P2] embryos effectively behaving as Exd mutants (**m**). Expression of the *S1+S2* fragment (**n**) or *S2* (**o**) in wild type embryos. **p** Scheme showing the *S1* and *S1+55* constructs. **a–c** highlight the segments where each Hox protein is expressed: Lab in the intercalary (ic); Dfd in the mandible (md) and maxilla (mx); Scr in the labium (lb) and the first thoracic (T1). Antp, Ubx, Abd-A and Abd-B are expressed in the (T1-A9) thoracic (T) and abdominal (A) trunk segments (only Ubx and Abd-A expression are shown in **c**). Embryos in **a–c** were expressing *UAS-lacZ* with the *arm-Gal4* line to label all cells in grey. In panels d-i the coloured horizontal bar represents the extension of the *S1* (grey), *S2* (green), and *S3* (orange) sequences. Scale bar 50 μm

embryos *vvl1+2* expression almost disappears from the maxillary and labial segments and is reduced in T2 through A7 segments (Fig. 1l, m). This result suggests that both monomer and Hox-cofactor sites are required for *vvl1+2* expression; activation in cephalic segments is strongly cofactor-dependent while expression in trunk segments is partially cofactor-independent, suggesting an interesting role for class 3 Hox-monomer binding sites.

**Contribution of *vvl1+2* fragments to overall expression**. To reduce the complexity of working with such a large number of predicted binding sites, we analysed the smaller fragments. It was previously shown that neither the *S1* nor the *S3* constructs drive reporter expression in isolation[16]. Accordingly, *S1+S2*, a reporter without *S3*, has a similar pattern of expression as the full *vvl1+2* (Fig. 1l, n). This focused our attention on two overlapping Hox-cofactor sites located in the S2 fragment (Fig. 1j and highlighted with asterisks in Fig. 1d–f). One, which we will refer to as the *main* site, is a canonical class 2 site (nTGATTAATnnn) whose core and flanking sequences have been perfectly conserved across at least 20 *Drosophila* species that diverged about 1 million years ago (Fig. 1j, blue). The other, which we will refer to as the *overlapping* site, is a class 3 core site whose core but not flanking sequence is conserved (Fig. 1j, red).

The S2 reporter drives an expression pattern similar to that of the *vvl1+2* and *S1-S2* constructs, although weaker and less defined, especially in the labial and T1 segments where expression is almost undetectable [Fig. 1o and ref. [16]]. The close to normal expression driven by S2 along T2-A8 is not surprising due to the predicted monomer Ubx binding sites and the *overlapping* class 3 site (Fig. 1f, i). However, the lack of expression in the labium and T1 segments was surprising considering the presence of the *main* class 2 site predicted to be bound by Scr in the labium and T1[11]. This observation made us suspect the existence of additional elements located in *S1* required for expression in the labial segment. To test this hypothesis we expanded the *S1* fragment, which by itself is inactive, adding the adjacent 55 bp containing the *S2 main* and *overlapping* sites (Fig. 1p). Interestingly, the extended *S1+55* reporter element drives expression in the maxilla and labium overlapping *vvl1+2* in these segments, but almost completely lacks T2-A8 expression (Fig. 2a). As expected for the presence of a class 2 Hox-cofactor binding site, expression driven by *S1+55* disappears in *Dfd Scr* double mutants and in *hth^{P2}* mutant embryos (Fig. 2b, c). Mutation of the *main* class 2 site in the *S1+55* reporter to a sequence predicted not to bind any Hox-cofactor combination, results in the loss of reporter expression (*S1+55MUT* reporter, Fig. 2t).

The above experiments show that *S1* is an inactive fragment of the *vvl1+2* enhancer that becomes active in the head segments when a class 2 specific binding site recognised by the Dfd and Scr proteins in complex with Exd-Hth cofactors is added. Curiously, they also show that the addition of the *overlapping* class 3 specific binding site to *S1* is not sufficient to drive strong expression in the trunk segments, consistent with the idea that Hox-monomer or additional Hox-Exd sites are also required.

**Hox-cofactor site affinity and spatial expression regulation**. To test the significance of the Hox-cofactor site affinity for *S1+55* spatial activation, we systematically changed the *main* binding site sequence based on our knowledge about the specificity of all Hox-cofactor complexes as derived from our previous in vitro SELEX-seq experiments[11]. We first mutated a single base in the core of the TGATTAAT class 2 *main* site to transform it to a TGATT**T**AT class 3 site. We find that this *S1+55cl3* mutated reporter, although still expressed in the labium and weakly in the

maxilla, now becomes activated in all trunk segments (Fig. 2d). In *Dfd Scr* double mutants *S1+55cl3* head expression disappears (see below), while in *Scr Antp Ubx* triple mutant embryos or in *hth^{P2}* embryos trunk expression disappears (Fig. 2e, f), showing that although it is expressed at lower levels than *vvl1+2* this reporter is now regulated by class 3 Hox-cofactor complexes. Thus, a single base change confers trunk Hox affinity and trunk expression even though the *S1+55cl3* fragment does not completely lose class 2 Hox-cofactor regulation.

We next tested the effect of mutating the *main* site in *S1+55* towards TGATT**GA**T to generate a class 1 Labial site. As described previously, *vvl1+2* is not activated anterior to the maxillary segment and neither is *S1+55* (Fig. 2a). Interestingly, in the *S1+55cl1* reporter, a more anterior patch of expression appears in the intercalary (ic) segment where the Lab Hox protein is expressed (Fig. 2g, h). This anterior intercalary segment patch disappears in *lab* or in *hth^{P2}* mutant embryos (Fig. 2i, j).

These results show that the rational modification of a single base pair in a Hox-cofactor core sequence can drastically modify the spatial regulation of a Hox target gene.

**Comparing in silico, in vitro and in vivo predictions**. The above results show that the Hox specificity predictions based on SELEX-seq data correlate well with the observed spatial expression patterns driven by the *S1+55* class 1, class 2 and class 3 variants. To further test this correlation, we extended the analysis to other variants by modifying the flanking sequences, which are predicted to increase class specific binding affinity. We compared the in silico predicted affinity for each site with the relative affinity measured in vitro by EMSA and with the expression pattern driven by these site modifications in whole embryos.

By EMSA, the Lab-cofactor complex bound weakly to oligos containing either a class 2 (*main*) or class 3 site and showed almost no binding to a mutant variant (Fig. 3i, colour code in lanes corresponds to the oligo sequences represented in panel 3 h). These results coincide with the observed lack of activation by Lab of the corresponding reporter genes in the intercalary segment (Fig. 3a, b, d). In contrast, the Lab-cofactor complex binds robustly to a class 1 site in EMSA (Fig. 3j) correlating with expression in the intercalary segment driven by *S1+55cl1*, which has a class 1 binding site (Fig. 2g and Fig. 3c arrows; see Fig. 3k for affinity predictions). Further, modification of the flanking sequences to generate a class 1 binding site with optimal flanks (cl1 OF), further increases binding in vitro (Fig. 3t–u) and increases expression in vivo (driven by the *S1+55cl1 OF* reporter gene; compare the intercalary patch in Fig. 2g with 2q) while still retaining a genetic requirement for *lab* and *hth* (Fig. 2r, s).

By EMSA, the Dfd-cofactor complex bound to a class 2 site, but also to class 3 and class 1 sites (Fig. 3l, m), consistent with a previously described binding promiscuity of class 2 proteins[11]. The only site not bound by Dfd-cofactor is the mutant site. These results fit with the levels of activation driven by the Dfd-cofactor complex in the maxilla and labium of the class 1–3 site variants (Fig. 3a–c), its failure to activate the reporter with the mutant binding site (Fig. 3d) and correlates with affinity predictions (Fig. 3n). Modification of the flanking sequences of this class 2 site to generate optimal flanks for the Dfd-cofactor complex (Fig. 3s), further increased binding in EMSA (Fig. 3r) and increased maxillary expression in embryos (Figs. 2k and 3e).

By EMSA, Ubx-cofactor complexes bound strongly to class 3 sites, and more weakly to class 1 and class 2 sites, and not at all to the mutant Hox site (Fig. 3o, p). These results fit with the lack of expression in the trunk of class 2 variants and the trunk expression driven by class 3 variants (Fig. 3a, b). However these results do not fit with the observed embryo trunk expression of

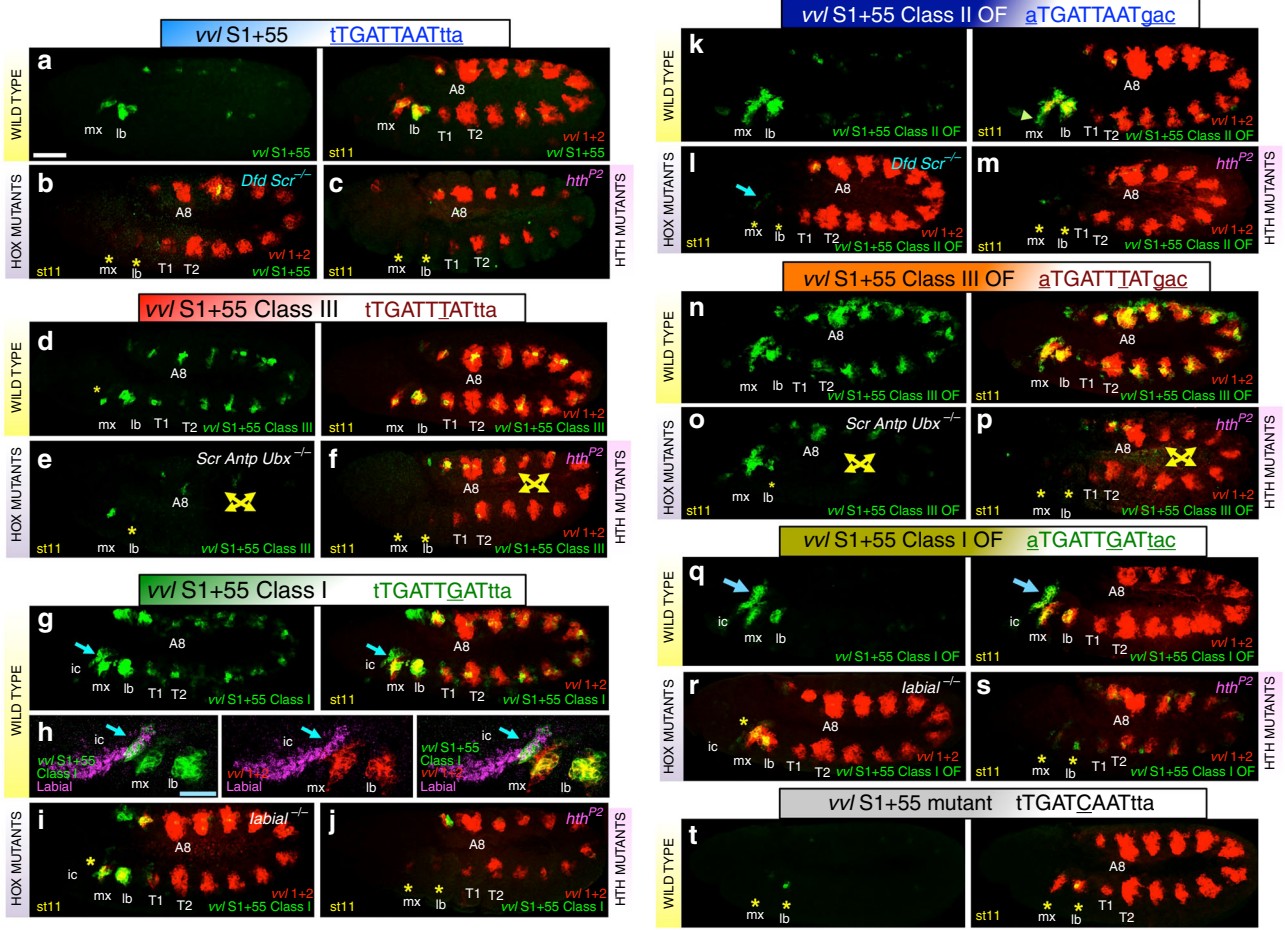

**Fig. 2** Expression of *S1+55* variants and *vvl1+2* in wild type and mutant backgrounds. Expression of *S1+55* class 2 reporter (**a**, left panel) compared to that of *vvl1+2* (**a**, right panel) in wild type, *Dfd Scr* (**b**) or *hth^P2* (**c**) mutant embryos. Asterisks mark the segments where *S1+55* expression is lost. Expression of *S1+55cl3* class 3 containing reporter (**d**, left panel) compared to that of *vvl1+2* in wild type (**d**, right panel), and its expression in *Scr Antp Ubx* (**e**) or *hth^P2* (**f**) mutant embryos. Yellow arrows mark the segments where *S1+55cl3* expression is lost. Note the maxillary expression in **e** is in the Dfd domain, which should not be affected in this genotype. Expression of *S1+55cl1* class 1 reporter (**g**, left panel) compared to that of *vvl1+2* in wild type (**g**, right panel). Blue arrow marks the additional intercalary (ic) segment patch where *vvl1+2* is not expressed. **h** Close-up of *S1+55cl1* expression (green) in the intercalary segment stained with anti-Lab antibody (purple) compared to *vvl1+2* (red) expression. Intercalary *S1+55cl1* expression is lost in *lab* mutant (**i**) or *hth^P2* (**j**) mutant embryos. **k** Expression of *S1+55cl2OF* class 2 optimal flanks reporter compared to that of *vvl1+2* in wild type (**k**, right panel); *Dfd Scr* (**l**) or *hth^P2* (**m**) mutant embryos. Arrowhead in (**k**, right panel) points to expression outside the normal *vvl1+2* domain. Asterisks in **l–m** mark the segments where *S1+55cl2OF* expression is lost. **n** Expression of *S1+55cl3OF* reporter compared to that of *vvl1+2* in wild type (**n**, right panel); *Scr Antp Ubx* (**o**) or *hth^P2* (**p**) mutant embryos. Yellow arrows mark the segments where *S1+55cl3OF* expression is lost. **q** Expression of *S1+55cl1OF* reporter compared to that of *vvl1+2* in wild type (**q**, right panel). Blue arrow marks the intercalary segment not expressing *vvl1+2* and exhibiting extended Lab expression. Intercalary *S1+55cl1OF* expression is lost in *lab* (**r**) or *hth^P2* (**s**) mutant embryos. Expression of *S1+55mut* (**t**), a Hox-cofactor mutant site reporter, compared to that of *vvl1+2* in wild type embryos (**t**, right panel). White scale bar 50 μm. Blue scale bar 25 μm. All embryos shown in these and following figures are at stage 11 (st11)

class 1 variants (Fig. 3c). This apparent inconsistency was resolved when we noticed that the class 1 mutation we had introduced, simultaneously modified the class 3 *overlapping* site into a high affinity class 3 site in the reporter constructs (Supplementary Fig. 1a). This high affinity site was not present in the oligos used in the EMSA experiments reported in Fig. 3. When we repeated the EMSA experiments with oligos that include this *overlapping* class 3 high affinity site, Ubx binding increased (Supplementary Fig. 1). These observations also agreed with our in silico predictions. Finally, as in our previous analyses, we modified the flanking sequences to generate a binding site with optimal flanks for Ubx and observed that this variant increased binding by EMSA (Fig. 3v, w) and increased expression in the trunk segments (Figs. 3g and 2n); this expression was still dependent on *Ubx* and *hth* (Fig. 2o, p). Notably, despite the strong class 3 OF binding site, the *S1+55cl3OF* drives weaker expression levels in the trunk compared to *vvl1+2* (Fig. 2n), which also includes many Hox-monomer binding sites.

These results show our in silico predictions match remarkably well with both the in vitro EMSA binding measurements and the in vivo expression patterns driven by different *S1+55* reporter variants.

**Conservation of Hox-cofactor affinity regulation.** Our results have demonstrated the specificity of Hox-cofactor DNA-binding affinity for controlling the spatial expression of its direct targets in *Drosophila*. Given the great conservation of the DNA-binding homeodomain and the cofactor-binding hexapeptide sequences in the orthologous Hox proteins of all animal species[20], we wondered if this regulatory mode is also conserved in the lineage that gave rise to vertebrates.

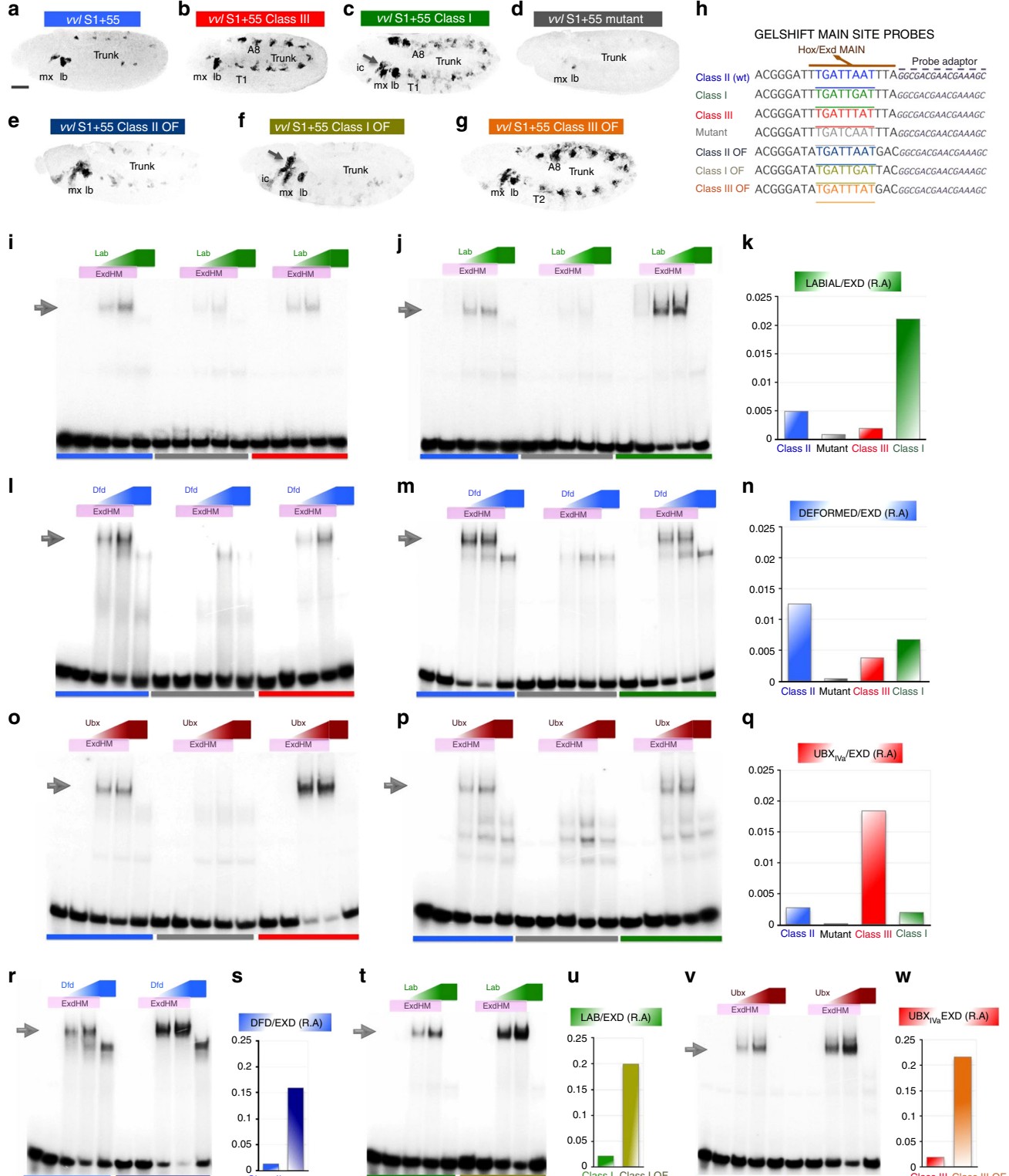

**Fig. 3** Embryonic expression compared with biochemical and in silico predicted site affinity. **a–g** Embryonic expression of the seven studied *S1+55* variants. **h** Sequences of the seven oligos tested by EMSA. **i–v** Hox-Exd and Hox-monomer affinity to the different oligos tested by EMSA. **i–j** Lab binding to class 2 site (blue lanes), mutant site (grey), class 3 site (red) or class 1 site (green). **k** In silico predicted affinity of Lab-Exd to the sites shown in **i–j**. **l–m** Dfd binding to class 2 site (blue), mutant site (grey), class 3 site (red) or class 1 site (green). **n** In silico predicted affinity of Dfd-Exd to the sites shown in **l–m**. **o–p** Ubx binding to class 2 site (blue), mutant site (grey), class 3 site (red) or class 1 site (green). **q** In silico predicted affinity of Ubx-Exd to the sites shown in **o–p**. **r** Dfd binding to class 2 site (blue) and class 2 optimal flanks site (dark blue). **s** In silico predicted affinity of Dfd-Exd to sites shown in **r**. **t** Lab binding to class 1 site (green) and class 1 optimal flanks site (olive green). **u** In silico predicted affinity of Lab-Exd to sites shown in **t**. **v** Ubx binding to class 3 site (red) and class 3 optimal flanks site (orange). **w** In silico predicted affinity of Ubx-Exd to sites shown in **v**. Scalebar 50 μm. Source data are provided as a Source Data file

To test this, we compared the activating capacity of *Drosophila* Hox proteins with that of the *Amphioxus lanceolatus* Hox orthologous proteins. As experiments with all *Drosophila* reporter lines gave consistent results (Fig. 2), we will only present in Fig. 4 illustrative examples of some of the most divergent DNA variants: *S1+55*, *S1+55cl2OF*, *S1+55cl1OF*, *S1+55cl3* and the endogenous *vvl1+2*. The experiments were performed by activating UAS-Hox expression with the *sal-Gal4* line in the maxillary and labial segments of *Dfd Scr* mutant embryos that lack endogenous Hox input and do not activate any of the reporter lines in these segments (see Fig. 4a–d asterisks). We chose to test *Amphioxus* proteins rather than other chordate's Hox such as chicken, mouse or human[21–23] because the Cephalocordata diverged before the two whole genome duplications that gave rise to four Hox clusters in vertebrates occurred, reducing the complexity of testing all duplicated paralogs with all the reporter variants[24–26].

We find that expression of *UAS-Dfd* and *UAS-AHox4*, which are the *Drosophila* and *Amphioxus* Hox orthologous genes, rescues to the same degree *S1+55cl2OF* and *vvl1+2* but only weakly *S1+55cl3* (compare Fig. 4e, f with g–h). Similarly, expression of *UAS-lab* and *UAS-AHox1* orthologs with *sal-Gal4*, rescues *S1+55cl1OF* but not the class 2 *S1+55* or *vvl1+2* (compare Fig. 4i, j with k–l). Finally, we find that *UAS-Ubx* and *UAS-AHox7* orthologs rescue *S1+55cl3* but not class 2 *S1+55* (compare Fig. 4m, n with o–p). These results are surprising given that the only conserved Hox sequences are the homeodomain and the hexapeptide (Supplementary Fig. 2), suggesting that the interaction with the collaborator proteins required for *vvl1+2* activation are either mediated by these conserved domains or do not require direct interaction with the Hox protein.

We conclude that the differential Hox-cofactor DNA-binding affinity strength has been conserved during evolution and that this mode of regulation is likely to be crucial for the spatial activation of direct Hox targets in all species. It will be interesting to test if all the different vertebrate paralogs of Hox1, Hox4 and Hox7 present in vertebrates maintain the same affinity-mediated regulatory potential in vivo.

**Importance of monomer Hox sites**. The above experiments tested the regulatory effect particular Hox-cofactor site variants impose on the inactive S1 fragment. To determine if the *main* and *overlapping* Hox-cofactor sites have the same influence on more robust enhancers, we analysed their contribution to the expression of the S2 fragment and the full *vvl1+2* element.

As our in silico analysis predicts that, besides the *main* and *overlapping* Hox-cofactor sites, the S2 fragment contains several putative monomer sites, we first studied the cofactor requirement for S2 activity. We find that in *hth[P2]* mutant embryos, expression driven by S2 is lost from all segments except A8, demonstrating a strong reliance on the Hox-cofactor sites (Fig. 5a, b). This result is confirmed by the similar expression levels we obtain after mutating in S2 a single base pair that simultaneously abolishes binding to both the *main* and *overlapping* Hox-cofactor sites (see *S2Hox/Exd main&OL mut*, Fig. 5c). Interestingly, expression in *S2Hox/Exd main&OL mut* disappears in *Abd-B[M1]* mutants (Fig. 5d), suggesting that Abd-B has a higher capacity to act through additional binding sites than other class 3 Hox proteins. To find out if the remaining expression in A8 may be controlled by Abd-B monomer function, we studied if *UAS-AbdBm* ectopic expression driven with *arm-Gal4* restores S2 activity in *hth[P2]* mutants. We observe Abd-B is able to restore head and trunk expression in the absence of functional cofactors, confirming Abd-B has a higher capacity to function as monomer than other Hox proteins (Fig. 5e).

To find out if chordate monomer proteins also present differential regulatory capacity in the absence of cofactor proteins, we tested in *hth[P2]* mutant embryos the response of *vvl1+2* and S2 constructs to expression of *UAS-AHox4* or *UAS-AHox7* expressed in the maxilla and labium using a *sal-Gal4* line. We observe that, as their *Drosophila* orthologous proteins, neither AHox4 nor AHox7 can activate S2 expression in the absence of cofactors (Supplementary Fig. 3a, b). In contrast, *vvl1+2* can be rescued by AHox7 but not by AHox4 (Supplementary Fig. 3a, b) indicating that, at least in *Drosophila*, chordate class 3 proteins may also have a higher capacity to act as monomers than class 2 proteins.

We next tested the effect of mutating the *main* and *overlapping* Hox-cofactor sites in the full *vvl1+2* context. Contrary to the strong effect such mutations have on *S1+55* and S2 reporters, in a *vvl1+2Hox/Exd main&OL mut* reporter the expression is only modestly reduced in all segments except in A8 that is unaffected (Fig. 5f). Interestingly, the remaining expression in *vvl1+2Hox/Exd main&OL mut* is very similar to that of *vvl1+2* in *hth[P2]* mutants [compare in Fig. 5 panels b (red) and f (green)], implying that this residual expression could be mediated by non-cofactor dependent monomer Hox binding sites. To test this idea, we analysed a reporter where, in addition to mutating the *main* and *overlapping* Hox-cofactor sites, we mutated about 18 putative core monomer Hox binding sites that reduce both the cofactor and global monomer input (Supplementary Figs. 4–7). In this reporter, which we call *vvl1+2vvm* (where *vvm* stands for *very, very mutated*), the predicted Hox input is strongly reduced and only feeble levels of expression remain compared to those in *vvl1+2* (Fig. 5g). Although in *vvl1+2vvm* we attempted to introduce the fewest number of base pair changes to eliminate Hox input, we cannot discard that we inadvertently mutated an unknown collaborator binding site also required for *vvl1+2* expression. To confirm that the decreased expression in *vvl1+2vvm* is not caused by the collateral mutation of unidentified collaborator binding sites, we restored the *main* and *overlapping* Hox-cofactor sites in the *vvl1+2 vvm* construct back to wild type. The resulting *vvl1+2vvm-R* reporter recovers expression levels similar to those present in *vvl1+2* (Fig. 5h). However, although the expression levels are similar, the *vvl1+2vvm-R* reporter activity now is strongly dependent on Hox cofactors as shown by the almost complete loss of activity in *hth[P2]* mutant embryos (Fig. 5i).

These results show that the activation of a Hox-regulated CRM can be mediated either by Hox-monomer binding sites or by the presence of a Hox-cofactor site.

**Significance of Hox-collaborator inputs**. Two signalling pathways are required for *vvl* regulation, the JAK/STAT pathway that acts as an activator and the WNT pathway that acts as a repressor[16,27].

In *Df(1)os1A* mutants, which delete the genes encoding the three JAK/STAT activating extracellular ligands Upd1, Upd2 and Upd3[28], the expression of the *vvl1+2* reporter is almost lost (Fig. 6a, b). There are three STAT binding sites in *vvl1+2*, one present in each of the three S1-3 fragments[16]. To find out if the *vvl1+2* variants described above are equally sensitive to collaborator input as *vvl1+2*, we studied their expression in *Df(1)os1A* mutants. In this mutant background the expression driven by *S1+55*, *S1+55cl1* and *S1+55cl3* embryos disappears almost completely (Fig. 6c–h) while in the *S1+55cl1OF* and *S1+55cl2OF* mutants some expression remains, particularly in the intercalary segment (Fig. 6i–l). These results show that JAK/STAT inputs positively into all of the tested variants, but the degree to which it is required depends on the strength of the Hox

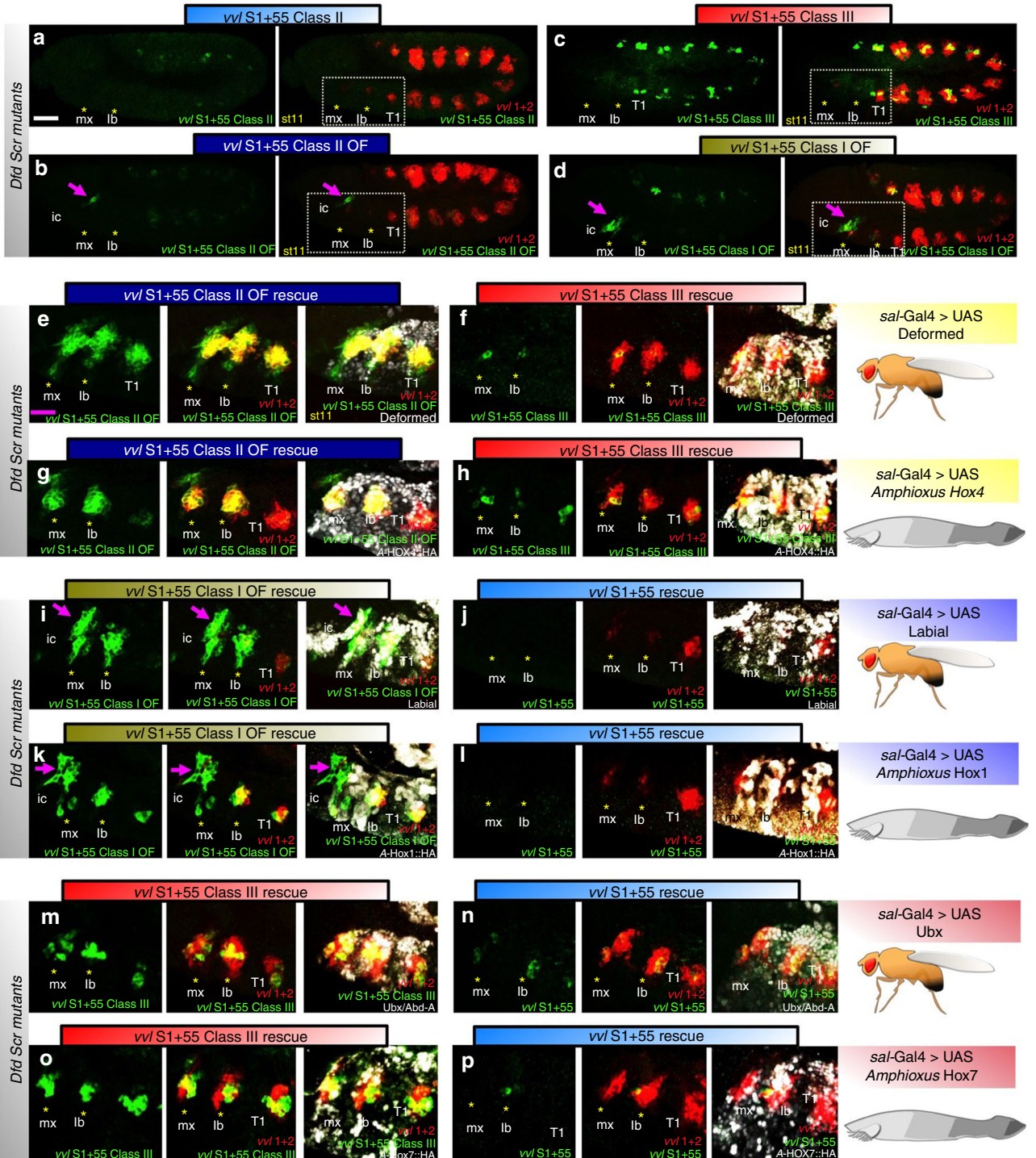

**Fig. 4** Sequence specific activation of Hox target genes by *Drosophila* and *Amphioxus* Hox proteins. Expression in *Dfd Scr* mutants of *S1+55* class 2 (**a**), *S1+55cl2OF* (**b**), *S1+55cl3* (**c**), *S1+55cl1OF* (**d**) is lost in the maxillary (mx) and labial (lb) segments (asterisks). Purple arrow points to intercalary segment expression. Rectangles in (a-d, right panels) highlight the intercalary (ic), maxillary (mx), labial (lb) and first thoracic segment (T1). No expression is observed in the mx or lb segments of *vvl1+2* (red) or any of the variants (asterisks), arrow points to ic segment expression. **e–p** Close-up of the ic, mx and lb segments of *Dfd Scr* embryos where a *Drosophila* or its *Amphioxus* orthologous Hox proteins are expressed to test their capacity to activate different *S1+55* variants. **e** Dfd can rescue *S1+55cl2OF* and *vvl1+2* but cannot activate *S1+55cl3* (**f**). **g** AHox4 can activate *S1+55cl2OF* but cannot activate *S1+55cl3* (**h**). **i** Lab can activate *S1+55cl1OF* but cannot activate *S1+55* containing a class 2 site (**j**). **k** AHox1 can activate *S1+55cl1OF* but cannot activate the *S1+55* containing a class 2site (**l**). **m** Ubx can rescue *S1+55cl3* but cannot activate *S1+55* containing a class 2 site (**n**). **o** Al-Hox7 can activate of *S1+55cl3* but cannot activate an *S1+55* class 2 containing site (**p**). White scale bar 50 μm, purple scale bar 25 μm

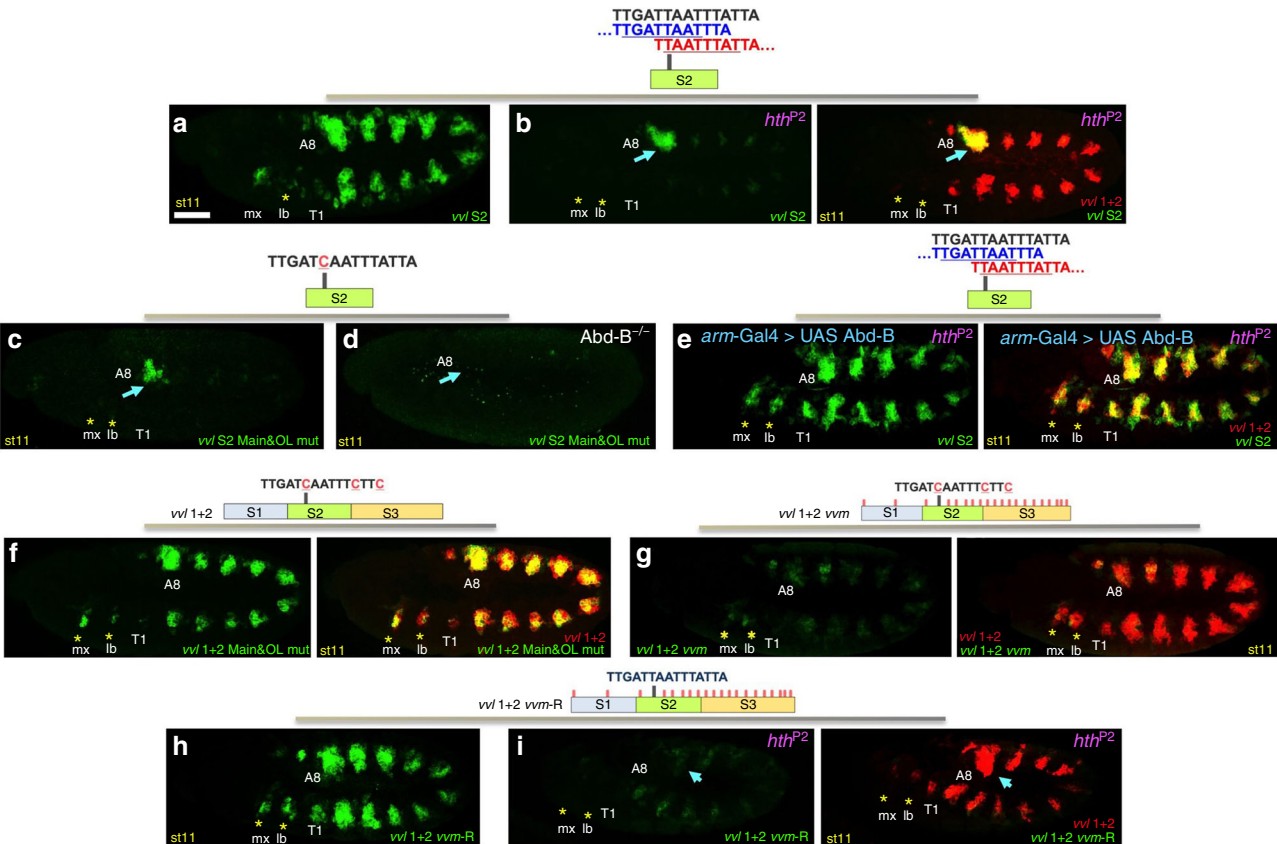

**Fig. 5** Relative requirement of Hox-cofactor vs. Hox-monomer sites in the *vvl1+2* and *S2* reporters. **a** Expression driven by the *S2* reporter (green) in wild type or in (**b**) *hth*[P2] embryos. (**b**, right panel) shows the differential effect absence of cofactor function causes in *S2* with respect to *vvl1+2* (red). Blue arrow points to A8 expression, which is maintained in both reporters, asterisks label the lack of expression in the maxilla and labial segments. *Main* and *overlapping* sites are present in both reporters. **c** The expression of *S2main&OL mut*, an *S2* variant with a single base change mutating both *main* and *overlapping* sites is restricted to the A8 segment. **d** In *Abd-B*[M1] homozygous embryos, the expression of the *S2main&OL mut* reporter disappears from A8. **e** The expression of *S2* in *hth*[P2] mutant embryos is restored by the ectopic Abd-B expression in the ectoderm to similar levels as those of *vvl1+2* (**e**, red in right panel). **f** The expression of *vvl1+2main&OL mut*, a *vvl1+2* reporter variant where the *main* and *overlapping* Hox-cofactor sites have been mutated, decreases in most segments, except A8, to similar levels as those observed in *vvl1+2 hth*[P2] mutant embryos. **g** Expression of the *vvl1+2vvm* reporter (green) where most Hox input is absent compared to *vvl1+2* (**g**, right panel). **h** Expression of the *vvl1+2vvm-R* reporter (green) where the *main* and *overlapping* Hox-cofactor sites have been restored. **i** Expression of the *vvl1+2vvm-R* reporter (green) in a *hth*[P2] mutant embryo showing the loss of most expression, including that in A8 (arrow) compared to the effects on *vvl1+2* (**i**, red in right panel)

input. Thus, reporter activity results from the additive inputs of both Hox and JAK/STAT.

In *wg* mutants the characteristic segmentally repeated patches of *vvl1+2* expression are replaced by a continuous stripe (Fig. 7a, b). To test if the normal and ectopic expression in these embryos requires Hox input, we analysed *wg Hox* mutant combinations. In *wg; Scr Antp Ubx* mutants the continuous *vvl1+2* stripe disappears from T1 to A1, while is maintained in A2 to A9 where *abd-A* and *Abd-B* are expressed (Fig. 7d). Analysis of *wg; Scr Antp Ubx abd-A* mutants shows that the *vvl1+2* reporter stripe is now missing from T1 to A7, while it is maintained in A8-A9 (and in two patches in A6-A7) where *Abd-B* is expressed (Fig. 7f). These experiments indicate that the WNT pathway induces effector proteins that block the capacity of the Hox proteins to activate *vvl1+2*.

The above results show that the embryo's Hox input is too weak to activate its direct target *vvl1+2* by itself and needs to be up-regulated by positive collaborator proteins to efficiently activate transcription. Similarly, the presence of negative collaborator proteins can down-regulate the Hox and STAT inputs, contributing to the final expression patterning. Thus, the equilibrium between weak Hox and STAT inputs with its negative collaborators defines the Hox target's expression.

## Discussion

Our knowledge on the regulation of direct Hox downstream genes has largely relied on the analysis of *cis*-regulatory modules controlled by a single Hox protein like the *fkh* salivary gland activation by Scr, the *ems* posterior spiracle activation by Abd-B, or the regulation of *svb* and *Dll* in the A1-A8 epithelium by Bithorax-Complex proteins[9,29–31]. Although these analyses reveal how individual Hox proteins, or small groups of Hox proteins, control particular targets, they have limited value for comparing the specificities of paralogous Hox proteins because each of this previously studied CRMs is activated in different cell types where they collaborate with different transcription factors and signalling pathways. In contrast, the analysis of the *vvl1+2* CRM offers a unique opportunity to directly compare how any Hox protein regulates the same target CRM, using the same collaborator inputs at homologous positions in all segments. This unique model not only provides a platform to compare between *Drosophila* Hox proteins but also between Hox proteins from divergent species that, in the case of Arthropods and Chordates, evolved independently for more that 500 million years.

One of the key findings of this work was the identification of the *S1* CRM fragment that cannot drive expression on its own,

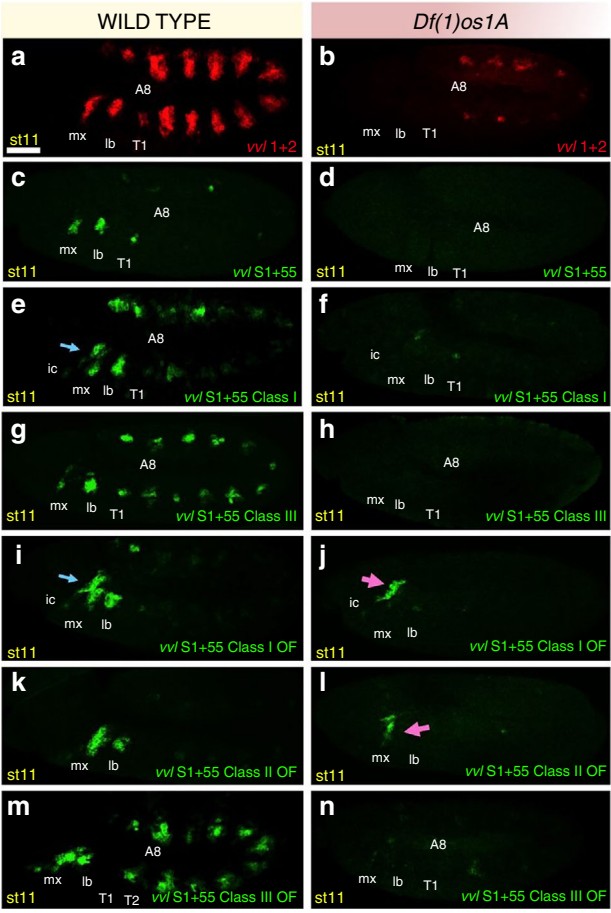

**Fig. 6** Requirement of JAK/STAT signalling for *vvl1+2* and *S1* variants. Expression of *vvl1+2* in wild type (**a**) and in *Df(1)os1A* embryos lacking all three JAK/STAT ligands (**b**). Expression of *S1+55* in wild type (**c**) or in *Df(1) os1A* embryos (**d**). Expression of *S1+55cl1* in wild type (**e**) or *Df(1)os1A* embryos (**f**). Expression of *S1+55cl3* in wild type (**g**) or *Df(1)os1A* embryos (**h**). Expression of *S1+55cl1OF* in wild type (**i**) or *Df(1)os1A* embryos (**j**). Expression of *S1+55cl2OF* in wild type (**k**) or *Df(1)os1A* embryos (**l**). Expression of *S1+55cl3OF* in wild type (**m**) or *Df(1)os1A* embryos (**n**)

but becomes active upon the addition of a small 55 bp element containing Hox-cofactor binding sites. This allowed us to test how small changes to the binding sites in the Hox-cofactor core sequence or in its flanks affect DNA binding affinity in vitro and the spatial expression of the *S1+55* variants in embryos. With this system we demonstrated that in silico predictions by *NRLB* were very consistent with in vitro biochemical affinity measurements and reporter gene analyses in vivo, making this algorithm a powerful tool to identify potential Hox-regulated sites in any organism.

Our results show that single base pair changes in the Hox-cofactor core binding sites are sufficient to modify their binding affinity to the Hox-cofactor complexes in vitro and, concomitantly, to alter the CRMs Hox-responsiveness in vivo. We also found that modifying base pairs that flank the core binding site can also affect Hox-cofactor affinity in vitro and activity in vivo, resulting in some cases in the reporter expanding its expression to cells where the wild type CRM is inactive. This observation indicates that, for some target CRMs, low affinity binding sites may be preferred over high affinity sites because they allow for further control of CRM activity by other inputs.

A particularly interesting observation was that *vvl1+2*, which as the endogenous *vvl* gene is not expressed in the intercalary

segment, can be activated in this anterior segment if the Hox-cofactor binding site is modified to a class 1 Lab binding site. This is a rare case where the mutation of a single DNA-binding site results in the ectopic expression of a target gene due to its recruitment of a new transcription factor. Thus, although the *Drosophila vvl* gene is expressed from the maxilla to the A9 segment, it has the potential to be expressed at homologous positions in all post-oral cephalic segments. It would be interesting to find out if other insect species express *vvl* in the intercalary segment, implying that the inability of Lab to regulate *vvl* in *Drosophila* may be a consequence of a loss of this regulation during evolution.

The DNA context in which a transcription factor binds can affect the way it regulates a downstream target gene. Such context, as defined by the surrounding DNA binding sites occupied by other factors, may favour or block the transcription factor's ability to recruit components of the basal transcription machinery, such as the mediator complex or the RNA polymerase, and will depend on the length and complexity of the adjacent accessible sequences. The *vvl* fragments we studied here vary in length from 200 to 680 bp. In these sequences we know of the existence of Hox and STAT binding sites and suspect of the presence of direct WNT mediator protein binding sites, although these last ones have not been identified biochemically. Besides these, currently unknown inputs may also contribute to the CRM's activity. Comparing the activity of *S1+55* with that of *S2*, where expression depends in both cases on the same Hox-cofactor binding sites, illustrates how different *cis*-regulatory contexts influence spatial expression. While the Hox-cofactor sites in *S1+55* exclusively activate cephalic expression, the same sites in *S2* can also activate trunk expression, demonstrating that regulation of Hox proteins is strongly influenced by neighbouring transcriptional inputs. Similarly, we find that in *S2*, the Hox-cofactor sites are absolutely required for expression in T1 to A7 while the same Hox-cofactor sites are dispensable for *vvl1+2* activity in those segments. Thus, the regulatory outcome of a Hox-cofactor binding site depends on the complexity of the CRM, with smaller CRMs being more sensitive to the DNA binding affinity of particular Hox-cofactor sites compared to larger elements that integrate several transcriptional inputs. The presence of these additional inputs may also help to give specificity to Hox class 2 proteins which as shown here and in Slattery et al., tend to be very promiscuous, binding also to class 1 and class 3 sites in vitro and in vivo.

Our results show that *vvl1+2* expression requires Dfd, Scr, Antp, Ubx, Abd-A and Abd-B (Fig. 7a, c, e and[14]). The analysis of *vvl1+2* expression in *hth^P2* embryos, where the Hox cofactors are not available, indicates that this requirement is mediated through both Hox-cofactor and Hox-monomer binding sites (Fig. 1m and Fig. 2c). The observation that mutation of the Hox-cofactor *main* and *overlapping* sites in the small *S1+55* or *S2* fragments abolishes almost completely their activity (Fig. 2t and Fig. 5c), while mutation of the same sites in *vvl1+2* has only a weak effect (Fig. 5f), points to additional sites acting in the *vvl1+2* context. The abundance of predicted monomer Hox binding sites in *S3* suggested that the differential behaviour between *S2* and *vvl1+2* could be mediated by the Hox-monomer sites. We confirmed this by showing that mutating in *vvl1+2vvm* 18 sites that in silico analyses predict to strongly decrease Hox-cofactor and monomer binding to the enhancer (Supplementary Figs. 4–7) nearly abolishes reporter expression (Fig. 5g). Interestingly, restoring the *main* and *overlapping* Hox-cofactor sites in this background restored a *vvl1+2*-like pattern of expression, showing that both types of binding sites can mediate the Ubx and Abd-A dependent activation (Fig. 5h).

The finding that in *hth^P2* embryos *S2* activity remains in the A8 segment, and that the same is true when the Hox-cofactor

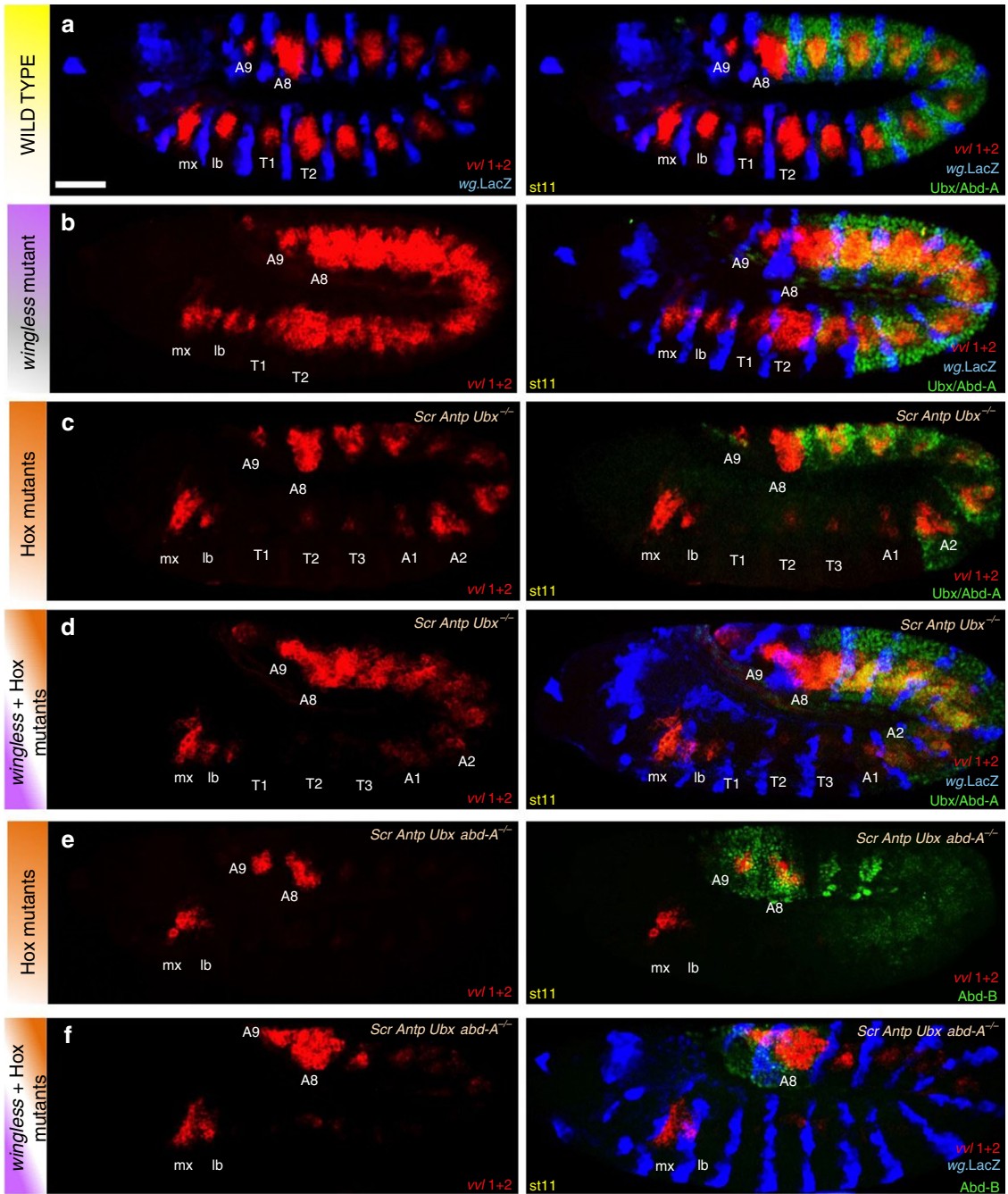

**Fig. 7** Requirement of WNT signalling and Hox input for *vvl1+2* activity. **a** Expression of *vvl1+2* (red) with respect to *wg* (blue), and with Ubx/Abd-A (green in right panel) in wild type embryos. Expression of *vvl1+2* (red) in *wg* mutant embryos (**b**) also stained to detect Ubx and Abd-A (green in right panel) and the *wg* expressing domain (blue). **c** Expression of *vvl1+2* (red) in *Scr Antp Ubx* mutant embryos with respect to Ubx/Abd-A (green in right panel). Expression of *vvl1+2* (red) in *wg; Scr Antp Ubx* mutant embryos (**d**) also stained to detect Ubx/Abd-A (green) and the *wg* expressing domain (blue in right panel). **e** Expression of *vvl1+2* (red) in *Scr Antp Ubx abd-A* mutant embryos with respect to Abd-B expression (green in right panel). Expression of *vvl1+2* (red) in *wg; Scr Antp Ubx abd-A* mutant embryos (**f**) also stained to detect Abd-B (green) and the *wg* expressing domain (blue in right panel)

sites are mutated in the *S2* reporters (Fig. 5b, c) plus the capacity of Abd-B to restore *S2* expression in *hth*[P2] embryos (Fig. 5e), shows that Abd-B can function as a monomer better than Ubx or Abd-A. This observation is consistent with previous reports showing that some Abd-B targets depend only on its monomer activity and that the presence of Hth-Exd cofactors can perturb their activation by Abd-B[31]. In the *vvl1+2* CRM studied here, Abd-B functions in A8 both through cofactor dependent (Fig. 5h, i) and independent binding sites (Fig. 5b, c).

Hox proteins have been shown to use negative collaborators to mediate some of their functions. For example, Ubx and Abd-A require the Engrailed (En) and Sloppy paired (Slp) proteins to repress the *Dll* leg CRM expression in the abdomen[30]. Here we describe another case of negative collaboration where the Hox positive input on the enhancer is blocked by the WNT pathway.

The confined expression of *vvl* into segmental patches is difficult to understand when we only take into account the capacity of all Hox proteins to activate *vvl1+2*. As every ectodermal cell in

the embryo expresses one or another Hox protein, if no other regulatory input existed, *vvl* should be expressed as a continuous stripe along the anterior posterior axis. Here we presented evidence that the expression of *vvl* in segmental patches is due to WNT downstream factors competing the Hox activating input. In the absence of WNT activity *vvl* is expressed as a continuous stripe due to the relief of this brake on Hox activity. The continuous stripe of *vvl1+2* disappears when both *wg* and *Hox* are mutant.

It will be interesting to find out how the WNT activated negative input blocks Hox activity on *vvl1+2* and if this mechanism is used to modulate other direct Hox targets.

Findings in this work lead us to conclude that when regulating their targets, Hox input can be mediated in three different ways.

First, the spatial expression where a target gene is activated may depend solely on the presence of high affinity Hox-cofactor binding sites if the CRM is small and does not contain repressor inputs. In cases where the affinity is high (like in the *S1+55* optimal flank binding sites we tested), or when several Hox-cofactor sites are multimerised [as when *fkh Scr* regulated sites were combined[32]] the target expression spreads within most of the segment.

Second, in cases where the Hox-cofactor affinity is low, the target activation may depend on the presence of positive collaborator sites (e.g. in *vvl1+2* through STAT binding sites). Interaction with collaborator proteins can also be negative as shown by the blocking effect the WNT pathway has on *vvl1+2* expression.

Third, Hox-monomer binding sites can be sufficient to compensate for the absence of high affinity Hox-cofactor binding sites [this report and[33]]. As exemplified by *vvl1+2*, these three modes of regulation can coexist with each other in a single CRM, fine-tuning its expression levels and pattern of expression.

The simplicity of the protein binding sites and their partial redundancy in a CRM can explain how minor mutations arising either in Hox or collaborator binding sites in a CRM may exist in a population with almost negligible effects on gene expression and without deleterious effects. Recombination of these polymorphisms can eventually result in mild target gene expression variations that could be selected for during evolution, gradually modifying the expression pattern of any Hox target.

In contrast to the potential for rapid changes in CRM activity, our results demonstrate that orthologous *Amphioxus* and *Drosophila* Hox proteins, which have evolved separately for 500 million years, have retained the ability to similarly activate CRMs with a wide range of Hox responsiveness, suggesting that the three modes of regulation we have described on *vvl1+2* may also hold true for gene regulation by vertebrate Hox proteins. We propose that the three modes of Hox target regulation we describe here exemplify how Hox proteins regulate their targets in the animal kingdom.

## Methods

**Fly Stocks**. The following mutant alleles and transgenic lines from the Bloomington *Drosophila* Stock Centre were used: *Dfd*[16]; *Scr*[4]; *lab*[14]; P{UAS-Dfd.B}W4; P{UAS-lab.M}X2;P{UAS-LacZ}; *Df(1)os1A*; *Scr*[C1] *Antp*[NS+RC3] *Ubx*[MX12] ref. [34], *Scr*[C1] *Antp*[NS+RC3] *Df109*, *hth*[P2], CyO*wg*[en11].

The following lines from our laboratory were used: UAS-Ubx; UAS-AbdBm; the reporter *vvl1+2* mCherry; the enhancer trap line *sal-GAL4* 459.2 and *arm*-Gal4.

**Enhancer-reporters**. All *cis*-regulatory modules used in this study were cloned into the EcoRI restriction site of the pCaSpeR-EGFP::PH plasmid in the same orientation[14] and transformed by the *Drosophila*-transformation platform using random P integration (CBM-SO, Universidad Autónoma de Madrid). To confirm the pattern of expression is independent of the insertion site, at least 4 different insertions were analysed for each construct.

The following *vvl1+2* enhancer variants driving membrane bound GFP-PH were generated by PCR: *S1+55*, *S2*, *S1+S2*. The different *S1+55* mutant versions,

as well as, *vvl1+2Hox/Exd main&OL mut* and *S2 Hox/Exd main&OL mut*, were generated by PCR mutagenesis. All mutations changing the *main* class 2 site simultaneously destroy the *overlapping* class 3 site except the class 1 mutant that created a strong class 3 site. The *vvl1+2vvm* construct bearing multiple mutations (Supplementary Fig. 8) was generated by chemical synthesis (GENEWIZ®). The *vvl1+2vvm* fragment was used as template to restore the Hox-Exd *main* and *overlapping* site by PCR mutagenesis, generating the *vvl1+2vvm-*R enhancer. The expression driven by all reporters was analysed in heterozygous embryos reared at 25 °C. To facilitate comparison between the different reporter lines we have summarised their expression in Supplementary Table 1.

A list of all primers used can be found as Supplementary Information primer list.

**pUASt *Amphioxus* Hox constructs**. Constructs to express C-terminal 3xHA-tagged *Amphioxus lanceolata* Al-HOX1, Al-HOX4 and Al-HOX7 proteins were generated by chemical synthesis (BIOMATIK) using, respectively, the following coding sequences (CDS): GenBank entries: EU921831, EU921832 and EU921834. The three gene synthesis products cloned into the pUASt vector were transformed into *Drosophila melanogaster* by the *Drosophila*-transformation platform selecting 3 random insertions (CBM-SO, Universidad Autónoma de Madrid).

**Inmunochemistry and microscopy**. *Drosophila* embryos raised at 25 °C were fixed during 20 min in a PBS (1×)-Formaldehyde 6% solution. Fixed embryos were washed twice in PBS (1×)-tween 0.2% and incubated during 4 h with primary antibodies diluted in a PBS (1×)-tween 0.2%-BSA 1% solution, washed for 1 h in a PBS (1×)-tween 0.2% solution and incubated 2 h with secondary antibodies diluted in PBS (1×)-tween 0.2%-BSA 1%. Finally, the embryo sample was washed in a PBS 1× solution for 1 h and mounted in Vectashield® (Vector laboratories).

The following primary antibodies were used: chicken anti-GFP (1:800; Abcam13970; lot.nr.: GR53074-3), rat anti-RFP (1:500; Chromotek 5F8; lot. nr.:140915), rabbit anti-β-Gal (1:1000; Cappel; lot.nr.:04623), mouse anti-β-Gal (1:1000; Pro-mega; lot.nr.:63535), rabbit anti-Dfd and anti-Labial (1:100; T. Kaufman), mouse anti-Ubx/abd-A FP6.87 (1:50; DSHB; lot.nr.:1ea 1/2/14), mouse anti-AbdB 1A2E9 (1:50; DSHB; lot.nr.: 1ea 5/14/15) rabbit anti-HA (1:500; Abcam 9110).

The following Invitrogen secondary antibodies were used: anti-chicken A488, anti-mouse A488, anti-mouse A555, anti-mouse A647, anti-rabbit A488, anti-rabbit A555, anti-rabbit A647 and anti-rat A555 (All diluted 1:200; Invitrogen).

Preparations were analysed using a Leica SPE confocal microscope. Confocal images were taken using a ×20/0.70 IMM and ×40/1.15 oil objectives. Confocal pictures were processed using ImageJ and Imaris (7.6).

**EMSAs: Hox protein purification and GelShifts**. His-tagged recombinant Lab (in vector pET14b), Dfd (in pET14b), and Ubx (isoform IVa; in pET21b) were expressed in BL21(DE3) cells (Agilent Technologies) through IPTG induction for ~4 h. Proteins were purified through Cobalt chromatography using TALON Metal Affinity Resin (Clontech, #635501). His-tagged HthHM (in pET21b) and non-tagged Exd (in pET9a) were co-expressed under the same conditions as the Hox recombinant proteins and they were co-purified as HM-Exd complex through binding to the TALON beads of the HthHM recombinant protein His-tag. Protein concentrations were determined by the Bradford assay and then confirmed by SDS/PAGE and Blue Coomassie analysis.

EMSA assays[35–38] were performed as follows. Double stranded 6 nM DNA probes labelled with [γ-$^{32}$P]ATP were used for the binding reactions. Exd-HM was used at a concentration of 200 nM in all cases. Labial, Deformed and UbxIVa concentrations ranged from 30 to 180 nM.

DNA-Protein binding reactions were loaded onto a 4% polyacrylamide gel. Samples were run for approximately 2.5 h at 120 V at 4 °C. Gels were vacuum dried and DNA binding was detected using phosphoRimaging. Images were taken using a Typhoon FLA 9500 scanner and processed using ImageJ (NIH).

Unprocessed scans of all EMSA experiments are included as supplementary information.

**Computational analysis**. Analysis of Exd-Hox and Hox-monomer binding sites in non-coding DNA sequence was performed using the binding affinity models reported in[12], using the R package NRLBtools available at github.com/BussemakerLab/NRLB. Further details will be provided upon request.

Hox protein sequence alignments were performed using ClustalW, UniProt and Jalview with default settings.

**Reporting summary**. Further information on research design is available in the Nature Research Reporting Summary linked to this article.

## Data availability

All relevant data supporting the key findings of this study are available within the article and its Supplementary Information files or from the corresponding authors upon reasonable request. A reporting summary for this Article is available as a Supplementary Information file.

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

## Acknowledgements

This work was supported by a María de Maeztu Unit of Excellence grant and a Ministerio de Innovación, Ciencia y Universidades grant cofunded by the European Regional Development Fund (FEDER) to JCGH, and grants from the National Institutes of Health to R.S.M. (R35 GM118336) and to H.J.B. (R01HG003008). C.S.H. received an EMBO short-term grant and a travel grant from The Company of Biologists. We thank E. Sánchez-Herrero for the critical reading of this paper and T. Kaufman for anti-Lab and anti-Dfd antibody.

## Author contributions

C.S.H. conceived the initial project and performed most experiments. C.S.H., R.M., and J.C.G.H. planned the experiments and wrote the paper. C.R. and H.J.B. helped to analyse biocomputationally the Hox predicted binding site affinity in the wild type and mutant *cis* regulatory modules. R.V. helped design the EMSA analysis.

## Additional information

**Competing interests:** The authors declare no competing interests.

