## [Peer Review File · Nature Communications]

Reviewers' comments:

Reviewer #1 (Remarks to the Author):

This manuscript by C. Sanchez-Higuera et al. addresses a longstanding and important question relative to the biology of HOX transcription factors: the "HOX paradox". The "HOX paradox" stands for the fact that HOX proteins fulfill specific roles in developing animals while sharing a conserved DNA-binding homeodomain thereby displaying very similar DNA binding specificities as monomers *in vitro*.

Starting from a cis-regulatory module (CRM) known to be the target of most HOX proteins in *Drosophila*, the authors explore how do HOX proteins specifically achieve their function by distinctly recognizing this shared responsive target.

By comparison with other studies in the field, the originality of the approach is to assemble all the experiments around a common and well-characterized target CRM and to combine *in vivo* (in *Drosophila* embryos) and *in vitro* (by gel shift assays) strategies to dissect the precise sequence requirements for the recognition and activation of this CRM. This in fact allows comparing with an unprecedented precision (at the nucleotide level) how different HOX proteins engage CRM activity.

The conclusions drawn by the authors consist in describing three modes of CRM binding which, differentially combined, subtend the selective mode of action of HOX proteins: as monomer, as HOX/TALE dimer, as well as by interacting with positive and negative modulators. Experiments also provide evidence that the rules applying to the functional specificity of *Drosophila* HOX proteins are evolutionary conserved since *Amphioxus* (chordate) proteins can substitute for *Drosophila* orthologues. Taken together the results presented in this work define a significant message to deliver to the "HOX community" and more largely to the fields of transcription and gene regulation in eukaryotes. It therefore deserves to appear in *Nature Comm*.

Nonetheless a few points could be raised, calling for modifications in the manuscript.

Major point.

The title ("Full spectrum *in vivo* Hox binding specificity..."), sentences in the abstract (e.g. : "Analysing a CRM directly regulated by seven different *Drosophila* Hox proteins, we uncover how Hox proteins..."), sentences in the introduction (e.g. "To systematically test the idea that...") are somewhat misleading because they suggest that the study have examined the mode of action of all *Drosophila* HOX proteins, which is not the case. Of course the data presented support that the conclusions drawn might be extrapolated to HOX proteins which have not been involved in the experiments (e.g. Scr, AntP, AbdA, AbdB not involved in experiments corresponding to EMSAs –Fig 3- or enhancer activation-Fig. 4). But formally the study is not exhaustive in the sense that all proteins have not been tested in all experiments. I would suggest to modify such misleading statements, including the title of the manuscript (for example remove "Full spectrum").

Other statements are somewhat too strong: e.g. in the introduction "Our study is the most comprehensive work to date on Hox target regulation and represents a paradigm for understanding how any other Hox target gene is controlled". Understanding "Hox target regulation" requires but cannot be limited to understanding how Hox proteins recognize enhancers to make them active. There are other molecular events taking place upon target regulation by a Hox protein (epigenetic changes, co-factor recruitment, polIII activation, etc...). Thus although quite significant the authors should not qualify their study as the "most comprehensive work to date on Hox target gene regulation". Along the same line, it is difficult to accept that, again although quite significant, this work will properly define "a paradigm for understanding how any other Hox target gene is controlled". Other scenarios than the three modalities of Hox target engagement highlighted in this study could formally be envisioned considering the vast repertoire of processes and targets into which Hox proteins are involved in the animal kingdom. I would suggest the authors to moderate such statements appearing in a few places in the manuscript.

Minor points

In the title and summary, I would suggest not to use an abbreviation like "CRM" or to define what "CRM" stands for.

Page 9: "In Df(1)os1A mutants, which delete the three JAK/STAT ligands...", I would suggest to clarify what "ligands" are for by writing "In Df(1)os1A mutants, which delete the three extracellular activators of the JAK/STAT pathway..."

Figure 1 K. It is indicated that the letter "W" stands for "A or T" in the depicted canonical consensus sequence, but this consensus does not contain any "W" letter.

Figure 3I; 3L; 3O. Schematic representations of "Exd-Lab-DNA" or "Lab-DNA" (Fig3I); "Exd-Dfd-DNA" or "Dfd-DNA" (Fig3L); "Exd-Ubx-DNA" (Fig3O) appear on the left side of EMSA gel pictures. What do these schemes indicate is not clear. If they indicate where protein-DNA complexes appear on the gels, it is confusing. Arrows or arrowheads would help instead. In addition, if these schemes indicate protein-DNA complexes in gels, it is problematic because a complex corresponding to "Lab-DNA" is not visible, neither on Fig 3I nor on Fig. 3J. Furthermore, such a schematic representation (with arrows for example) should appear next to other images (Fig. 3J, M, P, R, ...).

Figure 4E, a mention referring to "vvl S1+55 Class II OF" is missing (as appearing in green letters on images Fig. 4E', E").

Reference 17 appears incomplete (Journal name or book series missing).

Reviewer #2 (Remarks to the Author):

The manuscript by Carlos Sanchez-Higuera reports on a detailed enhancer analysis, providing clues to mechanisms of Hox transcription factor activity. The peculiarity of this study when compared to previous ones is that the focus is on an enhancer regulated by most *Drosophila* Hox proteins, (ie all but Labial). This provides an unprecedented opportunity to compare, within homologous cell clusters at different axial positions, molecular mechanisms of gene regulation by Hox transcription factors.

The study is of technical high quality and extremely well-executed and well-presented. It provides two main conclusions. The first one arises early in the manuscript (Fig.1), when it is shown that activation of the vvl1+2 enhancer (a 600bp element of the ventral veinless (vvl) gene) by *Drosophila* Hox proteins relies on distinct mechanisms : predominant binding of a Hox-cofactor (Hox-Exd) complex for class II Hox proteins (Dfd and Scr) in the anterior most region of the embryo, and predominant monomer binding (or at least Exd independent) for the class III Hox proteins (Antp->AbdB) in more posterior regions of the embryo. Distinct mechanisms for the regulation of a Distalless enhancer by *Drosophila* BX-C Hox proteins (Ubx/AbdA versus AbdB) was previously reported, but the present study provides a more extensive (Hox proteins from different classes) analysis than the previous ones, and thus provide a significant advance. Bases for the Hox cofactor versus Hox monomer regulatory mode could have been studied a little bit further (for example, but not only, as suggested in one of the specific points).

The second conclusion concerns the predictive value of our current and general understanding of Hox-cofactor DNA binding specificity. A very detailed analysis of a subfragment of the vvl1+2 enhancer (vvlS1+55; 200bp) is performed, using DNA mutations predicted to switch Hox class specificity. In vitro EMSA experiments and in vivo enhancer activity generally (also some problem may arise, see specific points) provide strong support for the current Hox-cofactor specificity model. It also provides further validation for a recently described in silico method, « NRLB » to predict transcription factor binding affinities. The work also establishes that *Amphioxus* orthologs of the *Drosophila* proteins display the same in vivo Hox-cofactor specificity. This by itself is important as only few such detailed analyses addressing the relevance of the Hox specificity predictive model are available. The finding that the effects of DNA mutations within the 200bp enhancer are different from those seen in the larger 600bp enhancer however questions the relevance of findings for the understanding of the pan-Hox regulation of the vvl gene. The examination of the role of additional and previously identified positive and negative regulatory

inputs, namely the activation by the JAK/STAT and repression by the WNT pathways, does not provide, as it stands, clues to understand the "enhancer context" specific effects.

Specific points

- The model of vvl enhancer regulation by Hox proteins does not consider the DNA binding activity of Hth. This could be probed by examining the expression of the vvl gene and activity of vvl enhancers in *Drosophila* Hth mutants that only produce a HD less Hth protein.
- If Hth does not participate to the Hox-Exd DNA binding complexes (as inferred from binding site labeling in the vvl enhancer), it is not clear why EMSA experiments are performed in the presence of Hth-HM.
- EMSA with *Amphioxus* proteins could be performed to parallel the in vivo work in order to ascertain that differences indeed rely on class specific Hox cofactor DNA binding activity.
- While de novo activity of the various vvl 200bp enhancers fits with model predictions, loss of activity, also predicted from the model, is often not observed. This needs discussion.
- It is stated that *Amphioxus* experiments with all reporter lines gave consistent results. It would then be more appropriate to show all data for a same category of modified enhancers (all OF enhancers for example) and also show the others as supplementary figures. Also the conclusion that the rescue is class specific could be strengthened by showing that it is less efficient when performed on vvl modified enhancer corresponding to the TWO other classes: for ex for Hox4 rescue, it is shown that rescue is efficient on a class II vvl enhancer, and not a class III enhancer. How Hox4 acts on a class I vvl enhancer is not presented. The same would apply for the Hox1 and Hox 7 rescue experiments.
- The authors could have taken advantages of the *Amphioxus* rescue experiments to address if the Hox-cofactor versus Hox monomer regulatory mechanisms are evolutionary conserved, performing rescue experiments in mutant contexts other than Dfd/Scr.
- It is stated several times in the manuscript that BX-C Hox proteins are closely related. While correct for Ubx/AbdA, this is not true for AbdB.

Reviewer #3 (Remarks to the Author):

The manuscript by Sanchez-Higueras et al presents the dissection of a regulatory module driving the expression of the vvl gene in *Drosophila* embryo, regulated by multiple different Hox genes in a direct manner. The main claims are (i) Hox factor regulates spatial expression in three modes (high affinity Hox-cofactor sites, through collaborator proteins, accumulation of Hox monomer binding sites ; (ii) these three modes exists in vertebrates and therefore across Metazoans.

Major comments:

The second claim is valid and well-supported. Regarding the third claim, the first two modes are well-supported by a series of in vivo reporter assays allowing to follow the spatial expression, as well as EMSA in vitro approaches. The third mode is less convincing from the presented data. The notion of "accumulation" is presented by the mutant with 18 monomer sites changed. But the effect of changing one or few monomers is not shown. Maybe one of these monomer site is actually of high affinity (matching fig11, where there is predictions of high-affinity and low-affinity monomer sites), and mutating all of them would reach the same phenotype. The claim for "accumulation" would thus need additional controls with individual mutations, or combinations of individual mutations, showing that the effect is only found when multiple monomer are involved.

The computational method section is not detailed enough to ensure reproducibility of the analyses. The input sequence must be described (genomic positions on assembly or provide the sequence itself as supplementary file) and the parameters used for NRLB mentioned (indicate "parameters used by default" if there was no tuning). It is unclear if NRLB uses a threshold and if all reported values above 0 on figure 1 should be considered as predictions, or if only predictions above a certain threshold should be considered. An integrated view of panels 1D-I would also be beneficial, with colors according to the Hox motifs. As such, it is difficult to assess if some sites overlap. If a same location is predicted for various motifs, it would become more evident that way (note that it is clearly explained that affinities should not be compared between motifs). Negative control are

also needed. What would be the predicted sites for the same sequence length, same composition ? In a ideal, it should be none, but this would provide a value for false positive predictions

The authors should help a bit more the reader to follow each steps of the numerous analyses of expression patterns, and facilitate the interpretation of the results. I strongly suggest to make a summary table having as rows each tested fragments (S1, S2, S3, S1+S2, S1+55, all variants...) and as columns (i) the contained Hox sites (class1, class2, monomers or Exd cofactor), (ii) the mutated factors if any, (iii) a summary of the observed expression : one column per expression domain (lab, mx, lb, T1....A9), with + and – signs in the cells of this part of the table to indicate if the expression is observed in each domain.

Some interpretation may be modulated:

p5: “consistent with the idea that Hox-monomer sites are also required”. It is unclear for me that it would necessarily be monomer sites. At the end of S2, there is a prediction of a Ubx/Exd site, so the interpretation would be that extra sites within S2 are required (either monomer or the predicted Ubx/Exd site).

p5 S1+55cl3: The weak expression in mx is lost in Dfd/Scr mutant (4C), but it is present in Scr/Antp/Ubx (2E), which should be indicated in the text. This suggests that the mx expression is due to Dfd.

p5 S1+55cl1: The more posterior expression in A9 (2G) is not described or discussed. This expression is not Lab dependent (2I), neither cofactor dependent. This seems to be related to the class3 overlapping site introduced with the class1 main. Additional experimental controls are needed, in particular in a Ubx mutant, not only Lab mutant.

p8: reference to Abd is not supported by in silico predictions or experiment in mutant. This part needs more data support to support the claims.

Minor comments:

- Fig 1J shows the conservation of the “main” Hox/Exd site. It would be interesting to have the information regarding cross-species conservation on all the Hox sites predicted, to assess if monomers are as conserved as the Hox/Exd in this CRM
- Public ChIP-seq datasets are not mentioned and/or used in complement of the reporter assays
- Figure 1D-I : precise in the legend that the horizontal bar above the positions represent the tested sequence + explain the color code
- The name “main” is somewhat misleading
- For clarity, precise in all figures that hthP2 is hth/Exd double mutant
- St11 in yellow is not explained in the legends
- “Sub-minimal S1 fragment” : naming is misleading, as the S1 is complete and completed with a S2 small fragment.

Manuscript NCOMMS-19-03458A

We thank the reviewers for their overall very positive comments (highlighted in green). Based on their comments, we have made important changes to the manuscript as detailed below. (Our answer are written in red, the reviewers' comments in black)

Reviewers' comments:

Reviewer #1 (Remarks to the Author):

This manuscript by C. Sanchez-Higuera et al. addresses a longstanding and important question relative to the biology of HOX transcription factors: the “HOX paradox”. The “HOX paradox” stands for the fact that HOX proteins fulfill specific roles in developing animals while sharing a conserved DNA-binding homeodomain thereby displaying very similar DNA binding specificities as monomers in vitro.

Starting from a cis-regulatory module (CRM) known to be the target of most HOX proteins in *Drosophila*, the authors explore how do HOX proteins specifically achieve their function by distinctly recognizing this shared responsive target.

By comparison with other studies in the field, the originality of the approach is to assemble all the experiments around a common and well-characterized target CRM and to combine in vivo (in *Drosophila* embryos) and in vitro (by gel shift assays) strategies to dissect the precise sequence requirements for the recognition and activation of this CRM. This in fact allows comparing with an unprecedented precision (at the nucleotide level) how different HOX proteins engage CRM activity.

The conclusions drawn by the authors consist in describing three modes of CRM binding which, differentially combined, subtend the selective mode of action of HOX proteins: as monomer, as HOX/TALE dimer, as well as by interacting with positive and negative modulators. Experiments also provide evidence that the rules applying to the functional specificity of *Drosophila* HOX proteins are evolutionary conserved since *Amphioxus* (chordate) proteins can substitute for *Drosophila* orthologues. Taken together the results presented in this work define a significant message to deliver to the “HOX community” and more largely to the fields of transcription and gene regulation in eukaryotes. It therefore deserves to appear in Nature Comm.

Nonetheless a few points could be raised, calling for modifications in the manuscript.

Major point.

The title (“Full spectrum in vivo Hox binding specificity...”), sentences in the abstract (e.g. : “Analysing a CRM directly regulated by seven different *Drosophila* Hox proteins, we uncover how Hox proteins...”), sentences in the introduction (e.g. “To systematically test the idea that...”) are somewhat misleading because they

suggest that the study have examined the mode of action of all Drosophila HOX proteins, which is not the case. Of course the data presented support that the conclusions drawn might be extrapolated to HOX proteins which have not been involved in the experiments (e.g. Scr, AntP, AbdA, AbdB not involved in experiments corresponding to EMSAs –Fig 3- or enhancer activation-Fig. 4). But formally the study is not exhaustive in the sense that all proteins have not been tested in all experiments. I would suggest to modify such misleading statements, including the title of the manuscript (for example remove “Full spectrum”).

Following the reviewer's suggestions, we have modified the title to:

***In vivo* Hox binding specificity revealed by systematic changes to a single cis regulatory module**

In the abstract we have modified the sentence:

“Analysing a CRM directly regulated by seven different Drosophila Hox proteins, we uncover how Hox proteins...”

to:

Analysing a cis regulatory module directly regulated by seven different Drosophila Hox proteins, we uncover how different Hox class proteins differentially control its expression

Other statements are somewhat too strong: e.g. in the introduction “Our study is the most comprehensive work to date on Hox target regulation and represents a paradigm for understanding how any other Hox target gene is controlled”. Understanding “Hox target regulation” requires but cannot be limited to understanding how Hox proteins recognize enhancers to make them active. There are other molecular events taking place upon target regulation by a Hox protein (epigenetic changes, co-factor recruitment, polII activation, etc...). Thus although quite significant the authors should not qualify their study as the “most comprehensive work to date on Hox target gene regulation”. Along the same line, it is difficult to accept that, again although quite significant, this work will properly define “a paradigm for understanding how any other Hox target gene is controlled”. Other scenarios than the three modalities of Hox target engagement highlighted in this study could formally be envisioned considering the vast repertoire of processes and targets into which Hox proteins are involved in the animal kingdom. I would suggest the authors to moderate such statements appearing in a few places in the manuscript.

We have modified the following sentence:

"Our study is the most comprehensive work to date on Hox target regulation and represents a paradigm for understanding how any other Hox target gene is controlled."

to:

By systematically modifying the nucleotide sequence of a direct Hox target, our study allows a direct comparison of Hox specificity at an unprecedented level of precision *in vivo*.

Minor points

In the title and summary, I would suggest not to use an abbreviation like “CRM” or to define what “CRM” stands for.

CRM has been replaced by **cis regulatory module** in title and abstract

Page 9: “In *Df(1)os1A* mutants, which delete the three JAK/STAT ligands...”, I would suggest to clarify what “ligands” are for by writing “In *Df(1)os1A* mutants, which delete the three extracellular activators of the JAK/STAT pathway...”

The sentence: In *Df(1)os1A* mutants, which delete the three JAK/STAT ligands has been changed to:

In *Df(1)os1A* mutants, which delete the genes encoding the three JAK/STAT activating extracellular ligands *Upd1*, *Upd2* and *Upd3*

Figure 1 K. It is indicated that the letter “W” stands for “A or T” in the depicted canonical consensus sequence, but this consensus does not contain any “W” letter.

We have deleted “W=A or T” from Fig 1K

Figure 3I; 3L; 3O. Schematic representations of “Exd-Lab-DNA” or “Lab-DNA” (Fig3I); “Exd-Dfd-DNA” or “Dfd-DNA” (Fig3L); “Exd-Ubx-DNA” (Fig3O) appear on the left side of EMSA gel pictures. What do these schemes indicate is not clear. If they indicate where protein-DNA complexes appear on the gels, it is confusing. Arrows or arrowheads would help instead. In addition, if these schemes indicate protein-DNA complexes in gels, it is problematic because a complex corresponding to “Lab-DNA” is not visible, neither on Fig 3I nor on Fig. 3J. Furthermore, such a schematic representation (with arrows for example) should appear next to other images (Fig. 3J, M, P, R, ...).

We have substituted the schemes for arrows and arrowheads in all gels.

Figure 4E, a mention referring to “*vvl S1+55 Class II OF*” is missing (as appearing in green letters on images Fig. 4E’, E”).

We have mentioned *vvl S1+55 Class II OF* in panel e.

Reference 17 appears incomplete (Journal name or book series missing).

Journal name has been included: *Current topics on Developmental Biology*

Reviewer #2 (Remarks to the Author):

The manuscript by Carlos Sanchez-Higueras reports on a detailed enhancer analysis, providing clues to mechanisms of Hox transcription factor activity. The peculiarity of this study when compared to previous ones is that the focus is on an enhancer regulated by most *Drosophila* Hox proteins, (ie all but Labial). This provides an

unprecedented opportunity to compare, within homologous cell clusters at different axial positions, molecular mechanisms of gene regulation by Hox transcription factors.

The study is of technical high quality and extremely well-executed and well-presented. It provides two main conclusions. The first one arises early in the manuscript (Fig.1), when it is shown that activation of the *vv1+2* enhancer (a 600bp element of the ventral veinless (*vv1*) gene) by Drosophila Hox proteins relies on distinct mechanisms : predominant binding of a Hox-cofactor (Hox-Exd) complex for class II Hox proteins (*Dfd* and *Scr*) in the anterior most region of the embryo, and predominant monomer binding (or at least Exd independent) for the class III Hox proteins (*Antp*->*AbdB*) in more posterior regions of the embryo. Distinct mechanisms for the regulation of a Distalless enhancer by Drosophila BX-C Hox proteins (*Ubx/AbdA* versus *AbdB*) was previously reported, but the present study provides a more extensive (Hox proteins from different classes) analysis than the previous ones, and thus provide a significant advance. Bases for the Hox cofactor versus Hox monomer regulatory mode could have been studied a little bit further (for example, but not only, as suggested in one of the specific points).

The second conclusion concerns the predictive value of our current and general understanding of Hox-cofactor DNA binding specificity. A very detailed analysis of a subfragment of the *vv1+2* enhancer (*vv1S1+55*; 200bp) is performed, using DNA mutations predicted to switch Hox class specificity. In vitro EMSA experiments and in vivo enhancer activity generally (also some problem may arise, see specific points) provide strong support for the current Hox-cofactor specificity model. It also provides further validation for a recently described in silico method, « NRLB » to predict transcription factor binding affinities. The work also establishes that *Amphioxus* orthologs of the Drosophila proteins display the same in vivo Hox-cofactor specificity. This by itself is important as only few such detailed analyses addressing the relevance of the Hox specificity predictive model are available. The finding that the effects of DNA mutations within the 200bp enhancer are different from those seen in the larger 600bp enhancer however questions the relevance of findings for the understanding of the pan-Hox regulation of the *vv1* gene.

We do not agree that this questions the relevance of our findings, although it does highlight the complexity that exists *in vivo*, namely, that including longer enhancer sequences is likely to increase the number of inputs it coordinates. Specifically, we show and discuss the idea that a short element containing a well-defined set of binding sites follows our Hox-Exd sequence specificity predictions very closely. In contrast, larger versions of the enhancer that contains additional monomeric binding sites add additional inputs that modify enhancer activity. In nature both types of inputs are likely to exist in different target genes. We speculate that the additional flexibility this adds to gene regulation may be key for evolution.

The examination of the role of additional and previously identified positive and negative regulatory inputs, namely the activation by the JAK/STAT and repression by the WNT pathways, does not provide, as it stands, clues to understand the “enhancer context” specific effects.

We do not agree. In particular, these results show that the spatial pattern of expression driven by Hox input can be significantly modulated by other transregulators and that the final spatial expression pattern cannot be understood without taking into account the additional inputs, which provide “context”.

Specific points

- The model of *vvl* enhancer regulation by Hox proteins does not consider the DNA binding activity of Hth. This could be probed by examining the expression of the *vvl* gene and activity of *vvl* enhancers in *Drosophila* Hth mutants that only produce a HD less Hth protein.

This and the following point are related.

This paper does not deal with the influence of Hth on Hox function, but on the function of the Hox-Exd complex. We use *hth* mutants because Hth is required for the nuclear import of Exd and it therefore provides a simple way to distinguish between activation mediated by Hox monomeric vs. Hox-Exd bound sites. The potential role for Hth binding is of course interesting, but beyond the scope of this paper. By focusing on the Hox-Exd binding site, we are able to be agnostic about a potential role for Hth binding, which should be unchanged in all of our manipulations.

- If Hth does not participate to the Hox-Exd DNA binding complexes (as inferred from binding site labeling in the *vvl* enhancer), it is not clear why EMSA experiments are performed in the presence of Hth-HM.

The EMSA experiments were done with HM-Exd-Hox for three reasons:

- 1) This complex naturally exists *in vivo*, whereas Exd-Hox complexes (without any Hth) do not, because Exd’s nuclear import depends on it being bound to Hth-HM.
- 2) The HM domain of Hth improves DNA binding by the Hox-Exd complex, presumably by altering the conformation of Exd.
- 3) The SELEX-seq data in which we base our experiments (Slattery et al *Cell* **147**, 1270-1282) were generated using Hth-HM-Exd-Hox complexes.

- EMSA with Amphioxus proteins could be performed to parallel the *in vivo* work in order to ascertain that differences indeed rely on class specific Hox cofactor DNA binding activity.

A SELEX-seq approach has been published with vertebrate proteins belonging to each of the three Hox classes, which show that the binding site preferences are the same as those of *Drosophila*.

Quoting Kribelbauer et al. Cell Reports 19:2383-95:

"Comparing the pattern of 12-bp oligomer enrichment from R0 to R1 for each

complex, we found similar cofactor-dependent differences in binding specificity between these Hox proteins, as previously observed for their D. melanogaster orthologs."

In our manuscript we chose to use *Amphioxus* Hox proteins because this species evolved before the chordate whole genome duplications occurred and each *Amphioxus* Hox protein is representative of a different vertebrate ortholog. Given the strong protein conservation at the level of homeodomain it is very unlikely that EMSA experiments with *Amphioxus* Hox proteins will differ from those with *Drosophila* Hox proteins. It is much more significant that the *Amphioxus* proteins work *in vivo*.

- While de novo activity of the various vvl 200bp enhancers fits with model predictions, loss of activity, also predicted from the model, is often not observed. This needs discussion.

In the text (pg6 regulation by Dfd) we explain that class 2 Hox proteins are more promiscuous being able to bind to class 3 and 1 sites, something already reported in (Slattery et al *Cell* **147**, 1270-1282):

"By EMSA, the Dfd-cofactor complex bound to a class 2 site, but also to class 3 and class 1 sites (Fig. 3L-M), consistent with a previously described binding promiscuity of class 2 proteins."

To clarify this, we have now also discussed the issue of class 2 Hox promiscuity in pg 12 where we say:

*"The presence of these additional inputs may also help to give specificity to Hox class 2 proteins which as shown here and in Slattery et al. tend to be very promiscuous, binding also to class 1 and class 3 sites *in vitro* and *in vivo*."*

- *It is stated that Amphioxus experiments with all reporter lines gave consistent results.*

We have to apologise. The sentence in the manuscript was incomplete, the word "*Drosophila*" was missing and as a result we have unwittingly confused the reviewer. The sentence, which we have now corrected in the manuscript (pg 7), should read as:

"As experiments with all Drosophila reporter lines gave consistent results (Fig.2), we will only present in Fig. 4 illustrative examples of some of the most divergent DNA variants: S1+55, S1+55cl20F, S1+55cl10F, S1+55cl3 and the endogenous vvl1+2."

Thus, we have not performed all 18 *Amphioxus* rescues but only the 6 we present in the figure. These were chosen because they give the clearest results as they belong to the three different Hox specificity classes. We believe that the similarity between the rescues obtained with the *Drosophila* and the orthologous *Amphioxus* Hox proteins is sufficient to make the conclusions in the paper.

-It would then be more appropriate to show all data for a same category of modified enhancers (all OF enhancers for example) and also show the others as supplementary figures. Also the conclusion that the rescue is class specific could be strengthened by showing that it is less efficient when performed on vvl modified

enhancer corresponding to the TWO other classes: for ex for Hox4 rescue, it is shown that rescue is efficient on a class II vvl enhancer, and not a class III enhancer. How Hox4 acts on a class I vvl enhancer is not presented. The same would apply for the Hox1 and Hox 7 rescue experiments.

See response to the previous point. We do not believe that doing the rescue experiments with all 18 reporter genes will significantly add to the conclusions already supported by the data in the paper.

- The authors could have taken advantages of the *Amphioxus* rescue experiments to address if the Hox-cofactor versus Hox monomer regulatory mechanisms are evolutionary conserved, performing rescue experiments in mutant contexts other than Dfd/Scr.

This is a very interesting point. Following the reviewer's suggestion we have tested the capacity of A-Hox4 and A-Hox7 to rescue *vvl1+2 mCherry* expression in *homothorax* mutants to see if these chordate Hox proteins can also function as monomers. We find that, in *Drosophila*, A-Hox 4 is unable to function in the absence of cofactor activity while A-Hox7 is capable. These results indicate that this *Amphioxus* class 3 protein is less dependent on cofactor presence than the class2 protein tested. We have described the experiment on pg 8 and include it as a new Supplementary Figure 3.

- It is stated several times in the manuscript that BX-C Hox proteins are closely related. While correct for Ubx/AbdA, this is not true for AbdB.

The reviewer is correct. Abd-B is one of the more divergent Hox proteins. We have now stated this point clearly through the text:

In introduction pg 2 we say:

"Moreover, with the exception of Abd-B which has one of the most divergent homeodomains and a preference for TTAT or TTAG sequences⁸, all Hox proteins bind the same TAAT core sequence⁷."

We have also deleted in discussion and introduction all instances where "closely related proteins" was applied to Bx-C proteins.

We have also introduced a couple of experiments dealing with the monomeric Abd-B specific regulation (see response to reviewer 3).

Reviewer #3 (Remarks to the Author):

The manuscript by Sanchez-Higuera et al presents the dissection of a regulatory module driving the expression of the *vvl* gene in *Drosophila* embryo, regulated by multiple different Hox genes in a direct manner. The main claims are (i) Hox factor regulates spatial expression in three modes (high affinity Hox-cofactor sites, through collaborator proteins, accumulation of Hox monomer binding sites ; (ii) these three modes exist in vertebrates and therefore across Metazoans.

Major comments:

The second claim is valid and well-supported. Regarding the third claim, the first two modes are well-supported by a series of in vivo reporter assays allowing to follow the spatial expression, as well as EMSA in vitro approaches. The third mode is less convincing from the presented data. The notion of “accumulation” is presented by the mutant with 18 monomer sites changed. But the effect of changing one or few monomers is not shown. Maybe one of these monomer site is actually of high affinity (matching fig1I, where there is predictions of high-affinity and low-affinity monomer sites), and mutating all of them would reach the same phenotype. The claim for “accumulation” would thus need additional controls with individual mutations, or combinations of individual mutations, showing that the effect is only found when multiple monomer are involved.

The reviewer is correct in saying that as we have not individually mutagenized each putative monomeric Hox site (there are about 18), it is formally possible that the enhancer regulation in *hth* mutants is not mediated by the cumulative binding of Hox to many monomer sites, but instead by a single monomer site. To correct this, we have changed the text avoiding saying " *accumulation of Hox-monomer binding sites* " and instead we say "*mediated by Hox-monomer sites*".

Although we believe that more than one Hox-monomer site is required, the presence of a high number of putative sites and the lack of sequence conservation in fragment S3 (Sotillos et al. 2010) containing most putative sites makes it difficult to identify the most important monomer site(s).

The computational method section is not detailed enough to ensure reproducibility of the analyses. The input sequence must be described (genomic positions on assembly or provide the sequence itself as supplementary file) and the parameters used for NRLB mentioned (indicate “parameters used by default” if there was no tuning). It is unclear if NRLB uses a threshold and if all reported values above 0 on figure 1 should be considered as predictions, or if only predictions above a certain threshold should be considered. An integrated view of panels 1D-I would also be beneficial, with colors according to the Hox motifs. As such, it is difficult to assess if some sites overlap. If a same location is predicted for various motifs, it would become more evident that way (note that it is clearly explained that affinities should not be compared between motifs). Negative control are also needed. What would be the predicted sites for the same sequence length, same composition ? In a ideal, it should be none, but this would provide a value for false positive predictions

We have included the *vv1+2* input sequence as Supplementary Fig. 8 where we compare its sequence with that of the *vv1+2vvm* reporter to facilitate understanding the mutations introduced.

We find combining panels D-I in Fig. 1, as suggested by the reviewer, is extremely confusing. Instead, we have included four new Supplementary Figures 4-7 that show the *vv1+2* predicted *in silico* binding site positions for either Lab, Dfd, Ubx or Abd-B as monomers or in combination with Exd. In these figures we have also presented the predicted binding sites in a *vv1+2vvm* reporter. We have traced a discontinuous

vertical line in the panels to facilitate the localization of the *main* and *overlapping* sites, labelling the main site with an asterisk and the overlapping site with a diamond.

All of the computational models and methods were published in Rastogi, C. *et al.* Accurate and sensitive quantification of protein-DNA binding affinity. *Proc Natl Acad Sci U S A* **115**, E3692-E3701, doi:10.1073/pnas.1714376115 (2018). which is cited in the manuscript; there shouldn't be any reason to republish those details. We have added in the Experimental Procedures section the following sentence:
"Further details will be provided upon request."

The authors should help a bit more the reader to follow each steps of the numerous analyses of expression patterns, and facilitate the interpretation of the results. I strongly suggest to make a summary table having as rows each tested fragments (S1, S2, S3, S1+S2, S1+55, all variants...) and as columns (i) the contained Hox sites (class1, class2, monomers or Exd cofactor), (ii) the mutated factors if any, (iii) a summary of the observed expression : one column per expression domain (lab, mx, lb, T1....A9), with + and - signs in the cells of this part of the table to indicate if the expression is observed in each domain.

As suggested by the reviewer, we have included a summary table. The following sentence has been included in Experimental Procedures at the end of the *Enhancer-reporters* section:
"*To facilitate comparison between the different reporter lines we have summarized their expression in Supplementary Table 1.*"

Some interpretation may be modulated:

p5: "consistent with the idea that Hox-monomer sites are also required". It is unclear for me that it would necessarily be monomer sites. At the end of S2, there is a prediction of a Ubx/Exd site, so the interpretation would be that extra sites within S2 are required (either monomer of the predicted Ubx/Exd site).

We have corrected the sentence to accommodate for the reviewer's criticism. The above mentioned sentence now reads: *..."consistent with the idea that Hox-monomer or additional Hox-Exd sites are also required."*

p5 S1+55cl3: The weak expression in mx is lost in Dfd/Scr mutant (4C), but it is present in Scr/Antp/Ubx (2E), which should be indicated in the text. This suggests that the mx expression is due to Dfd.

We have included the following sentence in Fig. 2 legend: "*Note the maxillary expression in (2e) is in the Dfd domain, which should not be affected in this genotype.*"

p5 S1+55cl1: The more posterior expression in A9 (2G) is not described or discussed. This expression is not Lab dependent (2I), neither cofactor dependent. This seems to be related to the class3 overlapping site introduced with the class1 main. Additional experimental controls are needed, in particular in a Ubx mutant, not only Lab mutant.

The expression the reviewer mentions may be mediated by a telson homeodomain expressed protein. In the past, it has been shown that cuticle cephalic-like structures appear in the telson of certain hox mutants (Lewis Nature (1978) 276:565; Struhl 1983 JEEM 76:297)). This could be a similar case. We do not believe this patch of expression should be discussed as it is out of the scope of the current manuscript.

p8: reference to Abd is not supported by in silico predictions or experiment in mutant. This part needs more data support to support the claims.

We have added two new experiments in Fig.5 to show the requirement of Abd-B. First, we show that the remaining expression in *vv1S2 Hox/Exd Main&OL* mut completely disappears in Abd-B homozygous embryos (New Fig. 5d). Second, we show that ectopic Abd-B expression in the ectoderm of *hth^{P2}* embryos results in ectopic patches of expression in more anterior segments, confirming that Abd-B is functional in the absence of cofactors (New Fig. 5e). We have also added an *in silico* prediction of Abd-B monomeric sites (Supplementary Fig. 7).

Minor comments:

- Fig 1J shows the conservation of the “main” Hox/Exd site. It would be interesting to have the information regarding cross-species conservation on all the Hox sites predicted, to assess if monomers are as conserved as the Hox/Exd in this CRM

The *vv1+2* sequence was published in Sotillos et al. 2010, where we reported the S3 region is not conserved. S3 is where the majority of the monomeric sites locate.

- Public ChIP-seq datasets are not mentioned and/or used in complement of the reporter assays

We don't believe mentioning the public ChIP datasets is necessary. Our experiments are sufficiently self-contained.

- Figure 1D-I : precise in the legend that the horizontal bar above the positions represent the tested sequence + explain the color code.

We have added the following sentence at the end of figure 1's legend: " In panels d-i the coloured horizontal bar represents the extension of the S1 (grey), S2 (green) and S3 (orange) sequences."

- The name “main” is somewhat misleading

We'd like to keep it. It is the main site we analyse in this study. We have considered alternative names but they are more misleading in different ways.

- For clarity, precise in all figures that hthP2 is hth/Exd double mutant

We have now clearly specified this in the main text, and have also mentioned it now in Fig. 1's legend.

- St11 in yellow is not explained in the legends

We have added the following sentence to Fig. 2 legend "*All embryos shown in these and following figures are at stage 11 (st11).*"

We have also explained in figure legends the scale bar sizes appearing in various figures.

- “Sub-minimal S1 fragment” : naming is misleading, as the S1 is complete and completed with a S2 small fragment.

We have changed it to "the inactive S1 fragment"

We hope that the above described changes will satisfy the reviewers.

REVIEWERS' COMMENTS:

Reviewer #1 (Remarks to the Author):

According to the reviewer's comments Sanchez-Higueras et al made significant changes to their manuscript now entitled "In vivo Hox binding specificity revealed by systematic changes to a single cis regulatory module".

The modifications introduced in the manuscript meet all the points I raised further to its initial submission.

As already mentioned in my first report, the results presented in this work define a significant message to deliver to the "HOX community" and more largely to the fields of transcription and gene regulation in eukaryotes. It therefore deserves to appear in Nature Comm.

Reviewer #2 (Remarks to the Author):

This revised manuscript either clarifies through changes in the text, or responds through experimental additions most of the point raised in the initial review, improving substantially the ms, which I find suited for publication in Nature communications.

There is one point, initially raised, regarding the contribution of Hth, that I suggest the authors could still consider addressing . Without engaging into a general study of the contribution of Hth to the regulation of the vvl gene, a focused addition addressing if the HM domain is sufficient for proper activity of the vvl 200 bp enhancer could be provided. Such an addition would improve the continuum between in vitro and in vivo experiments.

Reviewer #3 (Remarks to the Author):

The authors have fully addressed the points I raised in the first version.

Answers to reviewers:

All three reviewers consider the MS acceptable for publication at NatComms. However reviewer 2 still has a suggestion that we find beyond the scope of this paper. The experiment will not add much and it will delay unnecessarily its publication.

Reviewer #2 (Remarks to the Author):

This revised manuscript either clarifies through changes in the text, or responds through experimental additions most of the point raised in the initial review, improving substantially the ms, which I find suited for publication in Nature communications.

There is one point, initially raised, regarding the contribution of Hth, that I suggest the authors could still consider addressing . Without engaging into a general study of the contribution of Hth to the regulation of the vvl gene, a focused addition addressing if the HM domain is sufficient for proper activity of the vvl 200 bp enhancer could be provided. Such an addition would improve the continuum between in vitro and in vivo experiments.

Answer to Reviewer 2: We do not think the paper would benefit from adding the extra experiment reviewer 2 indicates. The experiment will not add much and it will delay unnecessarily its publication. As we already commented in our previous letter: *"This paper does not deal with the influence of Hth on Hox function, but on the function of the Hox-Exd complex. We use hth mutants because Hth is required for the nuclear import of Exd and it therefore provides a simple way to distinguish between activation mediated by Hox monomeric vs. Hox-Exd bound sites. The potential role for Hth binding is of course interesting, but beyond the scope of this paper."*

COMPLETE REVIEWERS' COMMENTS:

Reviewer #1 (Remarks to the Author):

According to the reviewer's comments Sanchez-Higuera et al made significant changes to their manuscript now entitled "In vivo Hox binding specificity revealed by systematic changes to a single cis regulatory module".

The modifications introduced in the manuscript meet all the points I raised further to its initial submission.

As already mentioned in my first report, the results presented in this work define a significant message to deliver to the "HOX community" and more largely to the fields of transcription and gene regulation in eukaryotes. It therefore deserves to appear in Nature Comm.

Reviewer #2 (Remarks to the Author):

This revised manuscript either clarifies through changes in the text, or responds through experimental additions most of the point raised in the initial review, improving substantially the ms, which I find suited for publication in Nature communications.

There is one point, initially raised, regarding the contribution of Hth, that I suggest the authors could still consider addressing . Without engaging into a general study of the contribution of Hth to the regulation of the vvl gene, a focused addition addressing if the HM domain is sufficient for proper activity of the vvl 200 bp enhancer could be provided. Such an addition would improve the continuum between in vitro and in vivo experiments.

Reviewer #3 (Remarks to the Author):

The authors have fully addressed the points I raised in the first version.